# A Baseline for Few-Shot Image Classification

**Guneet S. Dhillon**[1], **Pratik Chaudhari**[2*], **Avinash Ravichandran**[1], **Stefano Soatto**[1,3]
[1]Amazon Web Services, [2]University of Pennsylvania, [3]University of California, Los Angeles
{guneetsd, ravinash, soattos}@amazon.com, pratikac@seas.upenn.edu

## Abstract

Fine-tuning a deep network trained with the standard cross-entropy loss is a strong baseline for few-shot learning. When fine-tuned transductively, this outperforms the current state-of-the-art on standard datasets such as Mini-ImageNet, Tiered-ImageNet, CIFAR-FS and FC-100 with the same hyper-parameters. The simplicity of this approach enables us to demonstrate the first few-shot learning results on the ImageNet-21k dataset. We find that using a large number of meta-training classes results in high few-shot accuracies even for a large number of few-shot classes. We do not advocate our approach as *the* solution for few-shot learning, but simply use the results to highlight limitations of current benchmarks and few-shot protocols. We perform extensive studies on benchmark datasets to propose a metric that quantifies the "hardness" of a few-shot episode. This metric can be used to report the performance of few-shot algorithms in a more systematic way.

## 1 Introduction

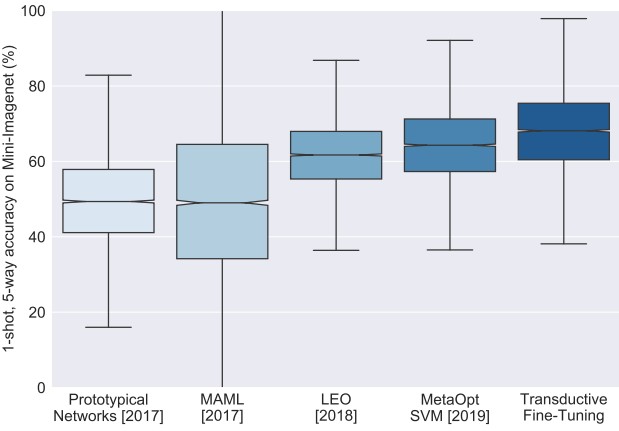

Figure 1: **Are we making progress?** The box-plot illustrates the performance of state-of-the-art few-shot algorithms on the Mini-ImageNet (Vinyals et al., 2016) dataset for the 1-shot 5-way protocol. The boxes show the $\pm$ 25% quantiles of the accuracy while the notches indicate the median and its 95% confidence interval. Whiskers denote the $1.5\times$ interquartile range which captures 99.3% of the probability mass for a normal distribution. The spread of the box-plots are large, indicating that the standard deviations of the few-shot accuracies is large too. This suggests that progress may be illusory, especially considering that none outperform the simple transductive fine-tuning baseline discussed in this paper (rightmost).

As image classification systems begin to tackle more and more classes, the cost of annotating a massive number of images and the difficulty of procuring images of rare categories increases. This has fueled interest in few-shot learning, where only few labeled samples per class are available for training. Fig. 1 displays a snapshot of the state-of-the-art. We estimated this plot by using published

---

*Work done while at Amazon Web Services

numbers for the estimate of the mean accuracy, the 95% confidence interval of this estimate and the number of few-shot episodes. For MAML (Finn et al., 2017) and MetaOpt SVM (Lee et al., 2019), we use the number of episodes in the author's Github implementation.

The field appears to be progressing steadily albeit slowly based on Fig. 1. However, the variance of the estimate of the mean accuracy is not the same as the variance of the accuracy. The former can be zero (e.g., asymptotically for an unbiased estimator), yet the latter could be arbitrarily large. The variance of the accuracies is extremely large in Fig. 1. This suggests that progress in the past few years may be less significant than it seems if one only looks at the mean accuracies. To compound the problem, many algorithms report results using different models for different number of ways (classes) and shots (number of labeled samples per class), with aggressive hyper-parameter optimization.[1] Our goal is to *develop a simple baseline for few-shot learning*, one that does not require specialized training depending on the number of ways or shots, nor hyper-parameter tuning for different protocols.

The simplest baseline we can think of is to pre-train a model on the meta-training dataset using the standard cross-entropy loss, and then fine-tune on the few-shot dataset. Although this approach is basic and has been considered before (Vinyals et al., 2016; Chen et al., 2018), it has gone unnoticed that it outperforms many sophisticated few-shot algorithms. Indeed, with a small twist of performing fine-tuning transductively, this baseline outperforms all state-of-the-art algorithms on all standard benchmarks and few-shot protocols (cf. Table 1).

Our contribution is to develop a *transductive fine-tuning* baseline for few-shot learning, our approach works even for a single labeled example and a single test datum per class. Our baseline outperforms the state-of-the-art on a variety of benchmark datasets such as Mini-ImageNet (Vinyals et al., 2016), Tiered-ImageNet (Ren et al., 2018), CIFAR-FS (Bertinetto et al., 2018) and FC-100 (Oreshkin et al., 2018), all with the same hyper-parameters. Current approaches to few-shot learning are hard to scale to large datasets. We report the first few-shot learning results on the ImageNet-21k dataset (Deng et al., 2009) which contains 14.2 million images across 21,814 classes. The rare classes in ImageNet-21k form a natural benchmark for few-shot learning.

The empirical performance of this baseline, should not be understood as us suggesting that this is *the* right way of performing few-shot learning. We believe that sophisticated meta-training, understanding taxonomies and meronomies, transfer learning, and domain adaptation are necessary for effective few-shot learning. The performance of the simple baseline however indicates that we need to interpret existing results[2] with a grain of salt, and be wary of methods that tailor to the benchmark. To facilitate that, we propose a *metric to quantify the hardness of few-shot episodes* and a way to systematically report performance for different few-shot protocols.

## 2   PROBLEM DEFINITION AND RELATED WORK

We first introduce some notation and formalize the few-shot image classification problem. Let $(x, y)$ denote an image and its ground-truth label respectively. The training and test datasets are $\mathcal{D}_s = \{(x_i, y_i)\}_{i=1}^{N_s}$ and $\mathcal{D}_q = \{(x_i, y_i)\}_{i=1}^{N_q}$ respectively, where $y_i \in C_t$ for some set of classes $C_t$. In the few-shot learning literature, training and test datasets are referred to as support and query datasets respectively, and are collectively called a few-shot episode. The number of *ways*, or classes, is $|C_t|$. The set $\{x_i \mid y_i = k, \ (x_i, y_i) \in \mathcal{D}_s\}$ is the *support* of class $k$ and its cardinality is $s$ *support shots* (this is non-zero and is generally shortened to *shots*). The number $s$ is small in the few-shot setting. The set $\{x_i \mid y_i = k, (x_i, y_i) \in \mathcal{D}_q\}$ is the *query* of class $k$ and its cardinality is $q$ *query shots*. The goal is to learn a function $F$ to exploit the training set $\mathcal{D}_s$ to predict the label of a test datum $x$,

---

[1]For instance, Rusu et al. (2018) tune for different few-shot protocols, with parameters changing by up to six orders of magnitude; Oreshkin et al. (2018) use a different query shot for different few-shot protocols.

[2]For instance, Vinyals et al. (2016); Ravi & Larochelle (2016) use different versions of Mini-ImageNet; Oreshkin et al. (2018) report results for meta-training on the training set while Qiao et al. (2018) use both the training and validation sets; Chen et al. (2018) use full-sized images from the parent ImageNet-1k dataset (Deng et al., 2009); Snell et al. (2017); Finn et al. (2017); Oreshkin et al. (2018); Rusu et al. (2018) use different model architectures of varying sizes, which makes it difficult to disentangle the effect of their algorithmic contributions.

where $(x, y) \in \mathcal{D}_q$, by

$$\hat{y} = F(x; \ \mathcal{D}_s). \tag{1}$$

Typical approaches for supervised learning replace $\mathcal{D}_s$ above with a statistic, $\theta^* = \theta^*(\mathcal{D}_s)$ that is, ideally, sufficient to classify $\mathcal{D}_s$, as measured by, say, the cross-entropy loss

$$\theta^*(\mathcal{D}_s) = \arg\min_{\theta} \frac{1}{N_s} \sum_{(x,y) \in \mathcal{D}_s} -\log p_\theta(y|x), \tag{2}$$

where $p_\theta(\cdot|x)$ is the probability distribution on $C_t$ as predicted by the model in response to input $x$. When presented with a test datum, the classification rule is typically chosen to be of the form

$$F_{\theta^*}(x; \mathcal{D}_s) \triangleq \arg\max_k p_{\theta^*}(k|x), \tag{3}$$

where $\mathcal{D}_s$ is *represented* by $\theta^*$. This form of the classifier entails a loss of generality *unless* $\theta^*$ is a sufficient statistic, $p_{\theta^*}(y|x) = p(y|x)$, which is of course never the case, especially given few labeled data in $\mathcal{D}_s$. However, it conveniently separates training and inference phases, never having to revisit the training set. This might be desirable in ordinary image classification, but not in few-shot learning. We therefore adopt the more general form of $F$ in (1).

If we call the test datum $x = x_{N_s+1}$, then we can obtain the general form of the classifier by

$$\hat{y} = F(x; \mathcal{D}_s) = \arg\min_{y_{N_s+1}} \min_{\theta} \frac{1}{N_s + 1} \sum_{i=1}^{N_s+1} -\log p_\theta(y_i|x_i). \tag{4}$$

In addition to the training set, one typically also has a *meta-training* set, $\mathcal{D}_m = \{(x_i, y_i)\}_{i=1}^{N_m}$, where $y_i \in C_m$, with set of classes $C_m$ disjoint from $C_t$. The goal of meta-training is to use $\mathcal{D}_m$ to infer the parameters of the few-shot learning model: $\hat{\theta}(\mathcal{D}_m; (\mathcal{D}_s, \mathcal{D}_q)) = \arg\min_{\theta} \frac{1}{N_m} \sum_{(x,y) \in \mathcal{D}_m} \ell(y, F_\theta(x; (\mathcal{D}_s, \mathcal{D}_q)))$, where meta-training loss $\ell$ depends on the method.

## 2.1 RELATED WORK

**Learning to learn:** The meta-training loss is designed to make few-shot training efficient (Utgoff, 1986; Schmidhuber, 1987; Baxter, 1995; Thrun, 1998). This approach partitions the problem into a base-level that performs standard supervised learning and a meta-level that accrues information from the base-level. Two main approaches have emerged to do so.

*Gradient-based approaches:* These approaches treat the updates of the base-level as a learnable mapping (Bengio et al., 1992). This mapping can be learnt using temporal models (Hochreiter et al., 2001; Ravi & Larochelle, 2016), or one can back-propagate the gradients across the base-level updates (Maclaurin et al., 2015; Finn et al., 2017). It is challenging to perform this dual or bi-level optimization, respectively. These approaches have not been shown to be competitive on large datasets. Recent approaches learn the base-level in closed-form using SVMs (Bertinetto et al., 2018; Lee et al., 2019) which restricts the capacity of the base-level although it alleviates the optimization problem.

*Metric-based approaches:* A majority of the state-of-the-art algorithms are metric-based approaches. These approaches learn an embedding that can be used to compare (Bromley et al., 1994; Chopra et al., 2005) or cluster (Vinyals et al., 2016; Snell et al., 2017) query samples. Recent approaches build upon this idea with increasing levels of sophistication in learning the embedding (Vinyals et al., 2016; Gidaris & Komodakis, 2018; Oreshkin et al., 2018), creating exemplars from the support set and picking a metric for the embedding (Gidaris & Komodakis, 2018; Allen et al., 2018; Ravichandran et al., 2019). There are numerous hyper-parameters involved in implementing these approaches which makes it hard to evaluate them systematically (Chen et al., 2018).

**Transductive learning:** This approach is more efficient at using few labeled data than supervised learning (Joachims, 1999; Zhou et al., 2004; Vapnik, 2013). The idea is to use information from the test datum $x$ to restrict the hypothesis space while searching for the classifier $F(x, \mathcal{D}_s)$ *at test time*. Our approach is closest to this line of work. We train a model on the meta-training set $\mathcal{D}_m$ and

initialize a classifier using the support set $\mathcal{D}_\mathrm{s}$. The parameters are then fine-tuned to adapt to the new test datum $x$.

There are recent papers in few-shot learning such as Nichol et al. (2018); Liu et al. (2018a) that are motivated from transductive learning and exploit the unlabeled query samples. The former updates batch-normalization parameters using query samples while the latter uses label propagation to estimate labels of all query samples at once.

**Semi-supervised learning:** We penalize the Shannon Entropy of the predictions on the query samples at test time. This is a simple technique in the semi-supervised learning literature, closest to Grandvalet & Bengio (2005). Modern augmentation techniques such as Miyato et al. (2015); Sajjadi et al. (2016); Dai et al. (2017) or graph-based approaches (Kipf & Welling, 2016) can also be used with our approach; we used the entropic penalty for the sake of simplicity.

Semi-supervised few-shot learning is typically formulated as having access to extra unlabeled data during meta-training or few-shot training (Garcia & Bruna, 2017; Ren et al., 2018). This is different from our approach which uses the unlabeled query samples for transductive learning.

**Initialization for fine-tuning:** We use recent ideas from the deep metric learning literature (Hu et al., 2015; Movshovitz-Attias et al., 2017; Qi et al., 2018; Chen et al., 2018; Gidaris & Komodakis, 2018) to initialize the meta-trained model for fine-tuning. These works connect the softmax cross-entropy loss with cosine distance and are discussed further in Section 3.1.

## 3 APPROACH

The simplest form of meta-training is pre-training with the cross-entropy loss, which yields

$$\hat{\theta} = \arg\min_\theta \; \frac{1}{N_\mathrm{m}} \sum_{(x,y)\in\mathcal{D}_\mathrm{m}} -\log p_\theta(y|x) + R(\theta), \tag{5}$$

where the second term denotes a regularizer, say weight decay $R(\theta) = \|\theta\|^2/2$. The model predicts logits $z_k(x;\theta)$ for $k \in C_\mathrm{m}$ and the distribution $p_\theta(\cdot|x)$ is computed from these logits using the softmax operator. This loss is typically minimized by stochastic gradient descent-based algorithms.

If few-shot training is performed according to the general form in (4), then the optimization is identical to that above and amounts to fine-tuning the pre-trained model. However, the model needs to be modified to account for the new classes. Careful initialization can make this process efficient.

### 3.1 SUPPORT-BASED INITIALIZATION

Given the pre-trained model (called the "backbone"), $p_\theta$ (dropping the hat from $\hat{\theta}$), we append a new fully-connected "classifier" layer that takes the logits of the backbone as input and predicts the labels in $C_\mathrm{t}$. For a support sample $(x, y)$, denote the logits of the backbone by $z(x; \theta) \in \mathbb{R}^{|C_\mathrm{m}|}$; the weights and biases of the classifier by $w \in \mathbb{R}^{|C_\mathrm{t}|\times|C_\mathrm{m}|}$ and $b \in \mathbb{R}^{|C_\mathrm{t}|}$ respectively; and the $k^\mathrm{th}$ row of $w$ and $b$ by $w_k$ and $b_k$ respectively. The ReLU non-linearity is denoted by $(\cdot)_+$.

If the classifier's logits are $z' = wz(x;\theta)_+ + b$, the first term in the cross-entropy loss: $-\log p_\Theta(y|x) = -w_y z(x;\theta)_+ - b_y + \log \sum_k e^{w_k z(x;\theta)_+ + b_k}$ would be the cosine distance between $w_y$ and $z(x;\theta)_+$ if both were normalized to unit $\ell_2$ norm and bias $b_y = 0$. This suggests

$$w_y = \frac{z(x;\theta)_+}{\|z(x;\theta)_+\|} \quad \text{and} \quad b_y = 0 \tag{6}$$

as a candidate for initializing the classifier, along with normalizing $z(x;\theta)_+$ to unit $\ell_2$ norm. It is easy to see that this maximizes the cosine similarity between features $z(x;\theta)_+$ and weights $w_y$. For multiple support samples per class, we take the Euclidean average of features $z(x;\theta)_+$ for each class in $C_\mathrm{t}$, before $\ell_2$ normalization in (6). The logits of the classifier are thus given by

$$\mathbb{R}^{|C_\mathrm{t}|} \ni z(x;\Theta) = w\frac{z(x;\theta)_+}{\|z(x;\theta)_+\|} + b, \tag{7}$$

where $\Theta = \{\theta, w, b\}$, the combined parameters of the backbone and the classifier. Note that we have added a ReLU non-linearity between the backbone and the classifier, *before* the $\ell_2$ normalization. All the parameters $\Theta$ are trainable in the fine-tuning phase.

**Remark 1 (Relation to weight imprinting).** The support-based initialization is motivated from previous papers (Hu et al., 2015; Movshovitz-Attias et al., 2017; Chen et al., 2018; Gidaris & Komodakis, 2018). In particular, Qi et al. (2018) use a similar technique, with minor differences, to expand the size of the final fully-connected layer (classifier) for low-shot continual learning. The authors call their technique "weight imprinting" because $w_k$ can be thought of as a template for class $k$. In our case, we are only interested in performing well on the few-shot classes.

**Remark 2 (Using logits of the backbone instead of features as input to the classifier).** A natural way to adapt the backbone to predict new classes is to re-initialize its final fully-connected layer (classifier). We instead append a new classifier after the logits of the backbone. This is motivated from Frosst et al. (2019) who show that for a trained backbone, outputs of all layers are entangled, without class-specific clusters; but the logits are peaked on the correct class, and are therefore well-clustered. The logits are thus better inputs to the classifier as compared to the features. We explore this choice via an experiment in Appendix C.6.

## 3.2    TRANSDUCTIVE FINE-TUNING

In (4), we assumed that there is a single query sample. However, we can also process multiple query samples together, and perform the minimization over all unknown query labels. We introduce a regularizer, similar to Grandvalet & Bengio (2005), as we seek outputs with a peaked posterior, or low Shannon Entropy $\mathbb{H}$. So the transductive fine-tuning phase solves for

$$\Theta^* = \arg\min_{\Theta} \frac{1}{N_{\mathrm{s}}} \sum_{(x,y) \in \mathcal{D}_{\mathrm{s}}} -\log p_\Theta(y \mid x) + \frac{1}{N_{\mathrm{q}}} \sum_{(x,y) \in \mathcal{D}_{\mathrm{q}}} \mathbb{H}(p_\Theta(\cdot \mid x)). \tag{8}$$

Note that the data fitting term uses the labeled support samples whereas the regularizer uses the unlabeled query samples. The two terms can be highly imbalanced (due to the varying range of values for the two quantities, or due to the variance in their estimates which depend on $N_{\mathrm{s}}$ and $N_{\mathrm{q}}$). To allow finer control on this imbalance, one can use a coefficient for the entropic term and/or a temperature in the softmax distribution of the query samples. Tuning these hyper-parameters per dataset and few-shot protocol leads to uniform improvements in the results in Section 4 by 1-2%. However, we wish to keep in line with our goal of developing a simple baseline and refrain from optimizing these hyper-parameters, and set them equal to 1 for all experiments on benchmark datasets.

## 4    EXPERIMENTAL RESULTS

We show results of transductive fine-tuning on benchmark datasets in few-shot learning, namely Mini-ImageNet (Vinyals et al., 2016), Tiered-ImageNet (Ren et al., 2018), CIFAR-FS (Bertinetto et al., 2018) and FC-100 (Oreshkin et al., 2018), in Section 4.1. We also show large-scale experiments on the ImageNet-21k dataset (Deng et al., 2009) in Section 4.2. Along with the analysis in Section 4.3, these help us design a metric that measures the hardness of an episode in Section 4.4. We sketch key points of the experimental setup here; see Appendix A for details.

**Pre-training:** We use the WRN-28-10 (Zagoruyko & Komodakis, 2016) model as the backbone. We pre-train using standard data augmentation, cross-entropy loss with label smoothing (Szegedy et al., 2016) of $\epsilon{=}0.1$, mixup regularization (Zhang et al., 2017) of $\alpha{=}0.25$, SGD with batch-size of 256, Nesterov's momentum of 0.9, weight-decay of $10^{-4}$ and no dropout. We use batch-normalization (Ioffe & Szegedy, 2015) but exclude its parameters from weight decay (Jia et al., 2018). We use cyclic learning rates (Smith, 2017) and half-precision distributed training on 8 GPUs (Howard et al., 2018) to reduce training time.

Each dataset has a training, validation and test set consisting of disjoint sets of classes. Some algorithms use only the training set as the meta-training set (Snell et al., 2017; Oreshkin et al., 2018), while others use both training and validation sets (Rusu et al., 2018). For completeness we report

results using both methodologies; the former is denoted as (train) while the latter is denoted as (train + val). All experiments in Sections 4.3 and 4.4 use the (train + val) setting.

**Fine-tuning:** We perform fine-tuning on one GPU in full-precision for 25 epochs and a fixed learning rate of $5 \times 10^{-5}$ with Adam (Kingma & Ba, 2014) without any regularization. We make two weight updates in each epoch: one for the cross-entropy term using support samples and one for the Shannon Entropy term using query samples (cf. (8)).

**Hyper-parameters:** We used images from ImageNet-1k belonging to the training classes of Mini-ImageNet as the validation set for pre-training the backbone for Mini-ImageNet. We used the validation set of Mini-ImageNet to choose hyper-parameters for fine-tuning. **All hyper-parameters are kept constant for experiments on benchmark datasets**.

**Evaluation:** Few-shot episodes contain classes sampled uniformly from classes in the test sets of the respective datasets; support and query samples are further sampled uniformly for each class; the query shot is fixed to 15 for all experiments unless noted otherwise. All networks are evaluated over 1,000 few-shot episodes unless noted otherwise. To enable easy comparison with existing literature, we report an estimate of the mean accuracy and the 95% confidence interval of this estimate. However, we encourage reporting the standard deviation in light of Section 1 and Fig. 1.

## 4.1 RESULTS ON BENCHMARK DATASETS

Table 1: **Few-shot accuracies on benchmark datasets for 5-way few-shot episodes.** The notation conv $(64^k)_{\times 4}$ denotes a CNN with 4 layers and $64^k$ channels in the $k^{\text{th}}$ layer. Best results in each column are shown in bold. Results where the support-based initialization is better than or comparable to existing algorithms are denoted by $\dagger$. The notation (train + val) indicates that the backbone was pre-trained on both training and validation sets of the datasets; the backbone is trained only on the training set otherwise. (Lee et al., 2019) uses a $1.25\times$ wider ResNet-12 which we denote as ResNet-12 $^*$.

| Algorithm | Architecture | Mini-ImageNet | | Tiered-ImageNet | | CIFAR-FS | | FC-100 | |
|---|---|---|---|---|---|---|---|---|---|
| | | 1-shot (%) | 5-shot (%) | 1-shot (%) | 5-shot (%) | 1-shot (%) | 5-shot (%) | 1-shot (%) | 5-shot (%) |
| Matching networks (Vinyals et al., 2016) | conv $(64)_{\times 4}$ | 46.6 | 60 | | | | | | |
| LSTM meta-learner (Ravi & Larochelle, 2016) | conv $(64)_{\times 4}$ | $43.44 \pm 0.77$ | $60.60 \pm 0.71$ | | | | | | |
| Prototypical Networks (Snell et al., 2017) | conv $(64)_{\times 4}$ | $49.42 \pm 0.78$ | $68.20 \pm 0.66$ | | | | | | |
| MAML (Finn et al., 2017) | conv $(32)_{\times 4}$ | $48.70 \pm 1.84$ | $63.11 \pm 0.92$ | | | | | | |
| R2D2 (Bertinetto et al., 2018) | conv $(96^k)_{\times 4}$ | $51.8 \pm 0.2$ | $68.4 \pm 0.2$ | | | $65.4 \pm 0.2$ | $79.4 \pm 0.2$ | | |
| TADAM (Oreshkin et al., 2018) | ResNet-12 | $58.5 \pm 0.3$ | $76.7 \pm 0.3$ | | | | | $40.1 \pm 0.4$ | $56.1 \pm 0.4$ |
| Transductive Propagation (Liu et al., 2018b) | conv $(64)_{\times 4}$ | $55.51 \pm 0.86$ | $69.86 \pm 0.65$ | $59.91 \pm 0.94$ | $73.30 \pm 0.75$ | | | | |
| Transductive Propagation (Liu et al., 2018b) | ResNet-12 | 59.46 | 75.64 | | | | | | |
| MetaOpt SVM (Lee et al., 2019) | ResNet-12 $^*$ | $62.64 \pm 0.61$ | $\mathbf{78.63 \pm 0.46}$ | $65.99 \pm 0.72$ | $81.56 \pm 0.53$ | $72.0 \pm 0.7$ | $84.2 \pm 0.5$ | $41.1 \pm 0.6$ | $55.5 \pm 0.6$ |
| Support-based initialization (train) | WRN-28-10 | $56.17 \pm 0.64$ | $73.31 \pm 0.53$ | $67.45 \pm 0.70^\dagger$ | $82.88 \pm 0.53^\dagger$ | $70.26 \pm 0.70$ | $83.82 \pm 0.49^\dagger$ | $36.82 \pm 0.51$ | $49.72 \pm 0.55$ |
| Fine-tuning (train) | WRN-28-10 | $57.73 \pm 0.62$ | $78.17 \pm 0.49$ | $66.58 \pm 0.70$ | $\mathbf{85.55 \pm 0.48}$ | $68.72 \pm 0.67$ | $\mathbf{86.11 \pm 0.47}$ | $38.25 \pm 0.52$ | $\mathbf{57.19 \pm 0.57}$ |
| Transductive fine-tuning (train) | WRN-28-10 | $\mathbf{65.73 \pm 0.68}$ | $78.40 \pm 0.52$ | $\mathbf{73.34 \pm 0.71}$ | $85.50 \pm 0.50$ | $\mathbf{76.58 \pm 0.68}$ | $85.79 \pm 0.50$ | $\mathbf{43.16 \pm 0.59}$ | $57.57 \pm 0.55$ |
| Activation to Parameter (Qiao et al., 2018) (train + val) | WRN-28-10 | $59.60 \pm 0.41$ | $73.74 \pm 0.19$ | | | | | | |
| LEO (Rusu et al., 2018) (train + val) | WRN-28-10 | $61.76 \pm 0.08$ | $77.59 \pm 0.12$ | $66.33 \pm 0.05$ | $81.44 \pm 0.09$ | | | | |
| MetaOpt SVM (Lee et al., 2019) (train + val) | ResNet-12 $^*$ | $64.09 \pm 0.62$ | $\mathbf{80.00 \pm 0.45}$ | $65.81 \pm 0.74$ | $81.75 \pm 0.53$ | $72.8 \pm 0.7$ | $85.0 \pm 0.5$ | $47.2 \pm 0.6$ | $62.5 \pm 0.6$ |
| Support-based initialization (train + val) | WRN-28-10 | $58.47 \pm 0.66$ | $75.56 \pm 0.52$ | $67.34 \pm 0.69^\dagger$ | $83.32 \pm 0.51^\dagger$ | $72.14 \pm 0.69^\dagger$ | $85.21 \pm 0.49^\dagger$ | $45.08 \pm 0.61$ | $60.05 \pm 0.60$ |
| Fine-tuning (train + val) | WRN-28-10 | $59.62 \pm 0.66$ | $\mathbf{79.93 \pm 0.47}$ | $66.23 \pm 0.68$ | $\mathbf{86.08 \pm 0.47}$ | $70.07 \pm 0.67$ | $\mathbf{87.26 \pm 0.45}$ | $43.80 \pm 0.58$ | $64.40 \pm 0.58$ |
| Transductive fine-tuning (train + val) | WRN-28-10 | $\mathbf{68.11 \pm 0.69}$ | $80.36 \pm 0.50$ | $\mathbf{72.87 \pm 0.71}$ | $86.15 \pm 0.50$ | $\mathbf{78.36 \pm 0.70}$ | $87.54 \pm 0.49$ | $\mathbf{50.44 \pm 0.68}$ | $\mathbf{65.74 \pm 0.60}$ |

Table 1 shows the results of transductive fine-tuning on benchmark datasets for standard few-shot protocols. We see that this simple baseline is uniformly better than state-of-the-art algorithms. We include results for support-based initialization, which does no fine-tuning; and for fine-tuning, which involves optimizing only the cross-entropy term in (8) using the labeled support samples.

The **support-based initialization is sometimes better than or comparable to state-of-the-art** algorithms (marked $\dagger$). The few-shot literature has gravitated towards larger backbones (Rusu et al., 2018). Our results indicate that for large backbones even standard cross-entropy pre-training and support-based initialization work well, similar to observation made by Chen et al. (2018).

For the 1-shot 5-way setting, fine-tuning using only the labeled support examples leads to minor improvement over the initialization, and sometimes marginal degradation. However, **for the 5-shot 5-way setting non-transductive fine-tuning is better than the state-of-the-art**.

In both (train) and (train + val) settings, **transductive fine-tuning leads to 2-7% improvement for 1-shot 5-way** setting over the state-of-the-art for all datasets. It results in an **increase of 1.5-4% for the 5-shot 5-way setting** except for the Mini-ImageNet dataset, where the performance is matched. This suggests that the **use of the unlabeled query samples is vital for the few-shot setting**.

For the Mini-ImageNet, CIFAR-FS and FC-100 datasets, using additional data from the validation set to pre-train the backbone results in 2-8% improvements; the improvement is smaller for Tiered-ImageNet. This suggests that **having more pre-training classes leads to improved few-shot performance** as a consequence of a better embedding. See Appendix C.5 for more experiments.

## 4.2 LARGE-SCALE FEW-SHOT LEARNING

The ImageNet-21k dataset (Deng et al., 2009) with 14.2M images across 21,814 classes is an ideal large-scale few-shot learning benchmark due to the high class imbalance. The simplicity of our approach allows us to present the first few-shot learning results on this large dataset. We use the 7,491 classes having more than 1,000 images each as the meta-training set and the next 13,007 classes with at least 10 images each for constructing few-shot episodes. See Appendix B for details.

Table 2: **Accuracy (%) on the few-shot data of ImageNet-21k.** The confidence intervals are large because we compute statistics only over 80 few-shot episodes so as to test for large number of ways.

| | | | Way | | | | | |
| Algorithm | Model | Shot | 5 | 10 | 20 | 40 | 80 | 160 |
| --- | --- | --- | --- | --- | --- | --- | --- | --- |
| Support-based initialization | WRN-28-10 | 1 | $87.20 \pm 1.72$ | $78.71 \pm 1.63$ | $69.48 \pm 1.30$ | $60.55 \pm 1.03$ | $49.15 \pm 0.68$ | $40.57 \pm 0.42$ |
| Transductive fine-tuning | WRN-28-10 | 1 | $89.00 \pm 1.86$ | $79.88 \pm 1.70$ | $69.66 \pm 1.30$ | $60.72 \pm 1.04$ | $48.88 \pm 0.66$ | $40.46 \pm 0.44$ |
| Support-based initialization | WRN-28-10 | 5 | $95.73 \pm 0.84$ | $91.00 \pm 1.09$ | $84.77 \pm 1.04$ | $78.10 \pm 0.79$ | $70.09 \pm 0.71$ | $61.93 \pm 0.45$ |
| Transductive fine-tuning | WRN-28-10 | 5 | $95.20 \pm 0.94$ | $90.61 \pm 1.03$ | $84.21 \pm 1.09$ | $77.13 \pm 0.82$ | $68.94 \pm 0.75$ | $60.11 \pm 0.48$ |

Table 2 shows the mean accuracy of transductive fine-tuning evaluated over 80 few-shot episodes on ImageNet-21k. The accuracy is extremely high as compared to corresponding results in Table 1 even for large way. E.g., the 1-shot 5-way accuracy on Tiered-ImageNet is $72.87 \pm 0.71\%$ while it is $89 \pm 1.86\%$ here. This corroborates the results in Section 4.1 and indicates that **pre-training with a large number of classes may be an effective strategy to build large-scale few-shot learning systems**.

The improvements of transductive fine-tuning are minor for ImageNet-21k because the support-based initialization accuracies are extremely high. We noticed a slight degradation of accuracies due to transductive fine-tuning at high ways because the entropic term in (8) is much larger than the the cross-entropy loss. The experiments for ImageNet-21k therefore scale down the entropic term by $\log |C_t|$ and forego the ReLU in (6) and (7). This reduces the difference in accuracies at high ways.

## 4.3 ANALYSIS

This section presents a comprehensive analysis of transductive fine-tuning on the Mini-ImageNet, Tiered-ImageNet and ImageNet-21k datasets.

**Robustness of transductive fine-tuning to query shot:** Fig. 2a shows the effect of changing the query shot on the mean accuracy. For the 1-shot 5-way setting, the entropic penalty in (8) helps as the query shot increases. This effect is minor in the 5-shot 5-way setting as more labeled data is available. Query shot of 1 achieves a relatively high mean accuracy because transductive fine-tuning can adapt to those few queries. **One query shot is enough to benefit from transductive fine-tuning**: for Mini-ImageNet, the 1-shot 5-way accuracy with query shot of 1 is $66.94 \pm 1.55\%$ which is better than non-transductive fine-tuning ($59.62 \pm 0.66\%$ in Table 1) and higher than other approaches.

**Performance for different way and support shot:** A few-shot system should be able to robustly handle different few-shot scenarios. Figs. 2b and 2c, show the performance of transductive fine-tuning

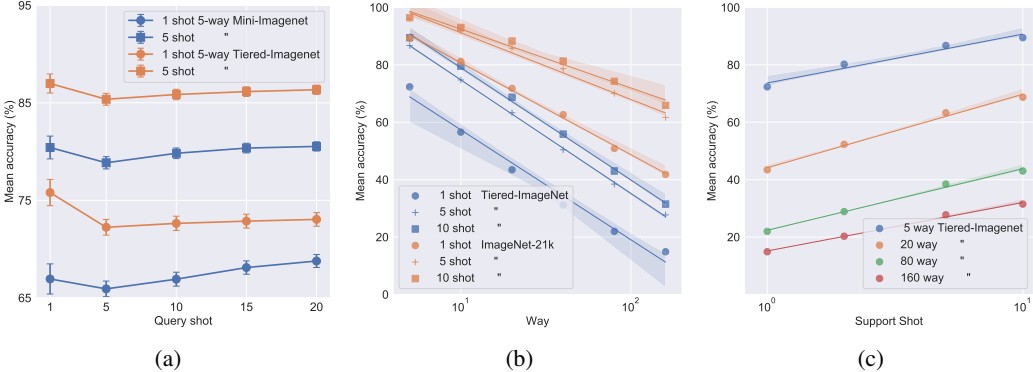

Figure 2: **Mean accuracy of transductive fine-tuning for different query shot, way and support shot.** Fig. 2a shows that the mean accuracy improves with query shot if the support shot is low; this effect is minor for Tiered-ImageNet. The mean accuracy for query shot of 1 is high because transductive fine-tuning can specialize to those queries. Fig. 2b shows that the mean accuracy degrades logarithmically with way for fixed support shot and query shot (15). Fig. 2c suggests that the mean accuracy improves logarithmically with the support shot for fixed way and query shot (15). These trends suggest thumb rules for building few-shot systems.

with changing way and support shot. **The mean accuracy changes logarithmically with the way and support shot** which provides thumb rules for building few-shot systems.

**Different backbone architectures:** We include experiments using conv $(64)_{\times 4}$ (Vinyals et al., 2016) and ResNet-12 (He et al., 2016a; Oreshkin et al., 2018) in Table 3, in order to facilitate comparisons for different backbone architectures. The results for transductive fine-tuning are comparable or better than state-of-the-art for a given backbone architecture, except for those in Liu et al. (2018b) who use a more sophisticated transductive algorithm using graph propagation, with conv $(64)_{\times 4}$. In line with our goal for simplicity, **we kept the hyper-parameters for pre-training and fine-tuning the same** as the ones used for WRN-28-10 (cf. Sections 3 and 4). These results show that **transductive fine-tuning is a sound baseline for a variety of backbone architectures**.

**Computational complexity:** There is no free lunch and our advocated baseline has its limitations. It performs gradient updates during the fine-tuning phase which makes it slow at inference time. Specifically, transductive fine-tuning is about $300\times$ slower (20.8 vs. 0.07 seconds) for a 1-shot 5-way episode with 15 query shot as compared to Snell et al. (2017) with the same backbone architecture (prototypical networks (Snell et al., 2017) do not update model parameters at inference time). The latency factor reduces with higher support shot. Interestingly, for a single query shot, the former takes 4 seconds vs. 0.07 seconds. This is a more reasonable factor of $50\times$, especially considering that the mean accuracy of the former is 66.2% compared to about 58% of the latter in our implementation. Experiments in Appendix C.3 suggest that using a smaller backbone architecture partially compensates for the latency with some degradation of accuracy. A number of approaches such as Ravi & Larochelle (2016); Finn et al. (2017); Rusu et al. (2018); Lee et al. (2019) also perform additional processing at inference time and are expected to be slow, along with other transductive approaches (Nichol et al., 2018; Liu et al., 2018b). Additionally, support-based initialization has the same inference time as Snell et al. (2017).

### 4.4 A PROPOSAL FOR REPORTING FEW-SHOT CLASSIFICATION PERFORMANCE

As discussed in Section 1, we need better metrics to report the performance of few-shot algorithms. There are two main issues: (i) standard deviation of the few-shot accuracy across different sampled episodes for a given algorithm, dataset and few-shot protocol is very high (cf. Fig. 1), and (ii) different models and hyper-parameters for different few-shot protocols makes evaluating algorithmic contributions difficult (cf. Table 1). This section takes a step towards resolving these issues.

**Hardness of an episode:** Classification performance on a few-shot episode is determined by the relative location of the features corresponding to labeled and unlabeled samples. If the unlabeled

features are close to the labeled features from the same class, the classifier can distinguish between the classes easily to obtain a high accuracy. Otherwise, the accuracy would be low. The following definition characterizes this intuition.

For training (support) set $\mathcal{D}_s$ and test (query) set $\mathcal{D}_q$, we will define the hardness $\Omega_\varphi$ as the average log-odds of a test datum being classified incorrectly. More precisely,

$$\Omega_\varphi(\mathcal{D}_q;\ \mathcal{D}_s) = \frac{1}{N_q} \sum_{(x,y)\in\mathcal{D}_q} \log \frac{1 - p(y\mid x)}{p(y\mid x)}, \tag{9}$$

where $p(\cdot\mid x)$ is a softmax distribution with logits $z_y = w\varphi(x)$. $w$ is the weight matrix constructed using (6) and $\mathcal{D}_s$; and $\varphi$ is the $\ell_2$ normalized logits computed using a rich-enough feature generator, say a deep network trained for standard image classification. This is a clustering loss where the labeled support samples form class-specific cluster centers. The cluster affinities are calculated using cosine-similarities, followed by the softmax operator to get the probability distribution $p(\cdot\mid x)$.

Note that $\Omega_\varphi$ does not depend on the few-shot learner and gives a measure of how difficult the classification problem is for any few-shot episode, using a generic feature extractor.

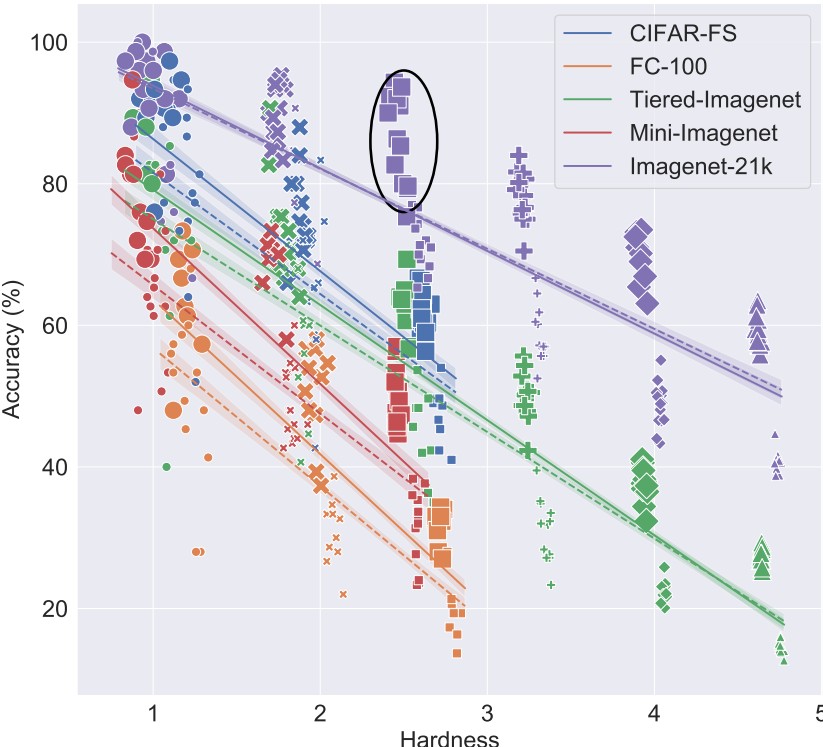

Figure 3: **Comparing the accuracy of transductive fine-tuning (solid lines) vs. support-based initialization (dotted lines) for different datasets, ways (5, 10, 20, 40, 80 and 160) and support shots (1 and 5).** Abscissae are computed using (9) and a Resnet-152 (He et al., 2016b) network trained for standard image classification on the ImageNet-1k dataset. Each marker indicates the accuracy of transductive fine-tuning on a few-shot episode; markers for support-based initialization are hidden to avoid clutter. Shape of the markers denotes different ways; ways increase from left to right (5, 10, 20, 40, 80 and 160). Size of the markers denotes different support shot (1 and 5); it increases from the bottom to the top. E.g., the ellipse contains accuracies of different 5-shot 10-way episodes for ImageNet-21k. Regression lines are drawn for each algorithm and dataset by combining the episodes of all few-shot protocols. This plot is akin to a precision-recall curve and allows comparing two algorithms for different few-shot scenarios. The areas in the first quadrant under the fitted regression lines are 295 vs. 284 (CIFAR-FS), 167 vs. 149 (FC-100), 208 vs. 194 (Mini-ImageNet), 280 vs. 270 (Tiered-ImageNet) and 475 vs. 484 (ImageNet-21k) for transductive fine-tuning and support-based initialization.

Fig. 3 demonstrates how to use the hardness metric. Few-shot **accuracy degrades linearly with hardness**. Performance for all hardness can thus be estimated by testing for two different ways. We advocate **selecting hyper-parameters using the area under the fitted curve as a metric** instead of tuning them specifically for each few-shot protocol. The advantage of such a test methodology is that it predicts the performance of the model across multiple few-shot protocols systematically.

**Different algorithms can be compared directly**, e.g., transductive fine-tuning (solid lines) and support-based initialization (dotted lines). For instance, the former leads to large improvements on easy episodes, the performance is similar for hard episodes, especially for Tiered-ImageNet and ImageNet-21k.

The **high standard deviation of accuracy** of few-shot learning algorithms in Fig. 1 can be seen as the spread of the cluster corresponding to each few-shot protocol, e.g., the ellipse in Fig. 3 denotes the 5-shot 10-way protocol for ImageNet-21k. It is the nature of few-shot learning that episodes have varying hardness even if the way and shot are fixed. However, episodes within the ellipse lie on a different line (with a large negative slope) which indicates that given a few-shot protocol, hardness is a good indicator of accuracy.

Fig. 3 also shows that due to fewer test classes, CIFAR-FS, FC-100 and Mini-ImageNet have less diversity in the hardness of episodes while Tiered-ImageNet and ImageNet-21k allow sampling of both very hard and very easy diverse episodes. For a given few-shot protocol, the hardness of episodes in the former three is almost the same as that of the latter two datasets. This indicates that CIFAR-FS, FC-100 and Mini-ImageNet may be good benchmarks for applications with few classes.

The hardness metric in (9) naturally builds upon existing ideas in deep metric learning (Qi et al., 2018). We propose it as a means to evaluate few-shot learning algorithms uniformly across different few-shot protocols for different datasets; ascertaining its efficacy and comparisons to other metrics will be part of future work.

## 5    DISCUSSION

Our aim is to provide grounding to the practice of few-shot learning. The current literature is in the spirit of increasingly sophisticated approaches for modest improvements in mean accuracy using an inadequate evaluation methodology. This is why we set out to establish a baseline, namely transductive fine-tuning, and a systematic evaluation methodology, namely the hardness metric. We would like to emphasize that our advocated baseline, namely transductive fine-tuning, is not novel and yet performs better than existing algorithms on all standard benchmarks. This is indeed surprising and indicates that we need to take a step back and re-evaluate the status quo in few-shot learning. We hope to use the results in this paper as guidelines for the development of new algorithms.

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

## A  SETUP

### A.1  DATASETS

We use the following datasets for our benchmarking experiments.

- The Mini-ImageNet dataset (Vinyals et al., 2016) which is a subset of ImageNet-1k (Deng et al., 2009) and consists of $84 \times 84$ sized images with 600 images per class. There are 64 training, 16 validation and 20 test classes. There are multiple versions of this dataset in the literature; we obtained the dataset from the authors of Gidaris & Komodakis (2018)[3].

- The Tiered-ImageNet dataset (Ren et al., 2018) is a larger subset of ImageNet-1k with 608 classes split as 351 training, 97 validation and 160 testing classes, each with about 1300 images of size $84 \times 84$. This dataset ensures that training, validation and test classes do not have a semantic overlap and is a potentially harder few-shot learning dataset.

- We also consider two smaller CIFAR-100 (Krizhevsky & Hinton, 2009) derivatives, both with $32 \times 32$ sized images and 600 images per class. The first is the CIFAR-FS dataset (Bertinetto et al., 2018) which splits classes randomly into 64 training, 16 validation and 20 test. The second is the FC-100 dataset (Oreshkin et al., 2018) which splits CIFAR-100 into 60 training, 20 validation and 20 test classes with minimal semantic overlap.

Each dataset has a training, validation and test set. The set of classes for each of these sets are disjoint from each other. For meta-training, we ran two sets of experiments: the first, where we only use the training set as the meta-training dataset, denoted by (train); the second, where we use both the training and validation sets as the meta-training dataset, denoted by (train + val). We use the test set to construct few-shot episodes.

### A.2  PRE-TRAINING

We use a wide residual network (Zagoruyko & Komodakis, 2016; Qiao et al., 2018; Rusu et al., 2018) with a widening factor of 10 and a depth of 28 which we denote as WRN-28-10. The smaller networks: conv $(64)_{\times 4}$ (Vinyals et al., 2016; Snell et al., 2017), ResNet-12 (He et al., 2016a; Oreshkin et al., 2018; Lee et al., 2019) and WRN-16-4 (Zagoruyko & Komodakis, 2016), are used for analysis in Appendix C. All networks are trained using SGD with a batch-size of 256, Nesterov's momentum set to 0.9, no dropout, weight decay of $10^{-4}$. We use batch-normalization (Ioffe & Szegedy, 2015). We use two-cycles of learning rate annealing (Smith, 2017), these are 40 and 80 epochs each for all datasets except ImageNet-21k, which uses cycles of 8 and 16 epochs each. The learning rate is set to $10^{-i}$ at the beginning of the $i^{\text{th}}$ cycle and decreased to $10^{-6}$ by the end of that cycle with a cosine schedule (Loshchilov & Hutter, 2016). We use data parallelism across 8 Nvidia V100 GPUs and half-precision training using techniques from Micikevicius et al. (2017); Howard et al. (2018).

We use the following regularization techniques that have been discovered in the non-few-shot, standard image classification literature (Xie et al., 2018) for pre-training the backbone.

- **Mixup (Zhang et al., 2017):** This augments data by a linear interpolation between input images and their one-hot labels. If $(x_1, y_1), (x_2, y_2) \in \mathcal{D}$ are two samples, mixup creates a new sample $(\tilde{x}, \tilde{y})$ where $\tilde{x} = \lambda x_1 + (1 - \lambda)x_2$ and its label $\tilde{y} = \lambda e_{y_1} + (1 - \lambda)e_{y_2}$; here $e_k$ is the one-hot vector with a non-zero $k^{\text{th}}$ entry and $\lambda \in [0, 1]$ is sampled from $\text{Beta}(\alpha, \alpha)$ for a hyper-parameter $\alpha$.

- **Label smoothing (Szegedy et al., 2016):** When using a softmax operator, the logits can increase or decrease in an unbounded manner causing numerical instabilities while training. Label smoothing sets $p_\theta(k|x) = 1 - \epsilon$ if $k = y$ and $\epsilon/(K - 1)$ otherwise, for a small constant $\epsilon > 0$ and number of classes $K$. The ratio between the largest and smallest output neuron is thus fixed which helps large-scale training.

- We exclude the batch-normalization parameters from weight-decay (Jia et al., 2018).

---

[3]https://github.com/gidariss/FewShotWithoutForgetting

We set $\epsilon = 0.1$ for label smoothing cross-entropy loss and $\alpha = 0.25$ for mixup regularization for all our experiments.

### A.3  FINE-TUNING HYPER-PARAMETERS

We used 1-shot 5-way episodes on the validation set of Mini-ImageNet to manually tune hyper-parameters. Fine-tuning is done for 25 epochs with a fixed learning rate of $5 \times 10^{-5}$ with Adam (Kingma & Ba, 2014). Adam is used here as it is more robust to large changes in the magnitude of the loss and gradients which occurs if the number of classes in the few-shot episode (ways) is large. We do not use any regularization (weight-decay, mixup, dropout, or label smoothing) in the fine-tuning phase. **These hyper-parameters are kept constant on all benchmark datasets**, namely Mini-ImageNet, Tiered-ImageNet, CIFAR-FS and FC-100.

All fine-tuning and evaluation is performed on a single GPU in full-precision. We update the parameters sequentially by computing the gradient of the two terms in (8) independently. This updates both the weights of the model and the batch-normalization parameters.

### A.4  DATA AUGMENTATION

Input images are normalized using the mean and standard-deviation computed on ImageNet-1k. Our Data augmentation consists of left-right flips with probability of 0.5, padding the image with 4px and adding brightness and contrast changes of $\pm$ 40%. The augmentation is kept the same for both pre-training and fine-tuning.

We explored augmentation using affine transforms of the images but found that adding this has minor effect with no particular trend on the numerical results.

### A.5  EVALUATION PROCEDURE

The few-shot episode contains classes that are uniformly sampled from the test classes of corresponding datasets. Support and query samples are further uniformly sampled for each class. The query shot is fixed to 15 for all experiments unless noted otherwise. We evaluate all networks over 1,000 episodes unless noted otherwise. For ease of comparison, we report the mean accuracy and the 95% confidence interval of the estimate of the mean accuracy.

## B  SETUP FOR IMAGENET-21K

The ImageNet-21k dataset (Deng et al., 2009) has 14.2M images across 21,814 classes. The blue region in Fig. 4 denotes our meta-training set with 7,491 classes, each with more than 1,000 images. The green region shows 13,007 classes with at least 10 images each, the set used to construct few-shot episodes. We do not use the red region consisting of 1,343 classes with less than 10 images each. We train the same backbone (WRN-28-10) with the same procedure as that in Appendix A on 84 $\times$ 84 resized images, albeit for only 24 epochs. Since we use the same hyper-parameters as the other benchmark datasets, we did not create validation sets for pre-training or the fine-tuning phases. The few-shot episodes are constructed in the same way as Appendix A. We evaluate using fewer few-shot episodes (80) on this dataset because we would like to demonstrate the performance across a large number of different ways.

## C  ADDITIONAL ANALYSIS

This section contains additional experiments and analysis, complementing Section 4.3. All experiments use the (train + val) setting, pre-training on both the training and validation sets of the corresponding datasets, unless noted otherwise.

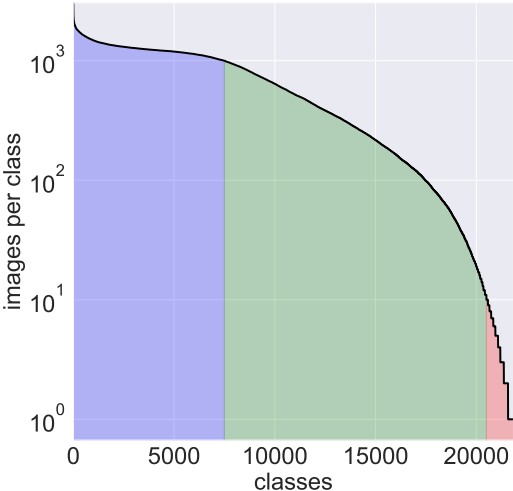

Figure 4: **ImageNet-21k is a highly imbalanced dataset.** The most frequent class has about 3K images while the rarest class has a single image.

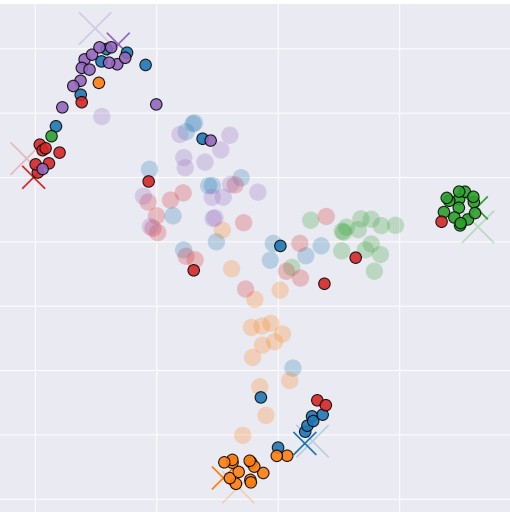

Figure 5: **t-SNE (Maaten & Hinton, 2008) embedding of the logits for 1-shot 5-way few-shot episode of Mini-ImageNet.** Colors denote the ground-truth labels; crosses denote the support samples; circles denote the query samples; translucent markers and opaque markers denote the embeddings before and after transductive fine-tuning respectively. Even though query samples are far away from their respective supports in the beginning, they move towards the supports by the end of transductive fine-tuning. Logits of support samples are relatively unchanged which suggests that the support-based initialization is effective.

## C.1 TRANSDUCTIVE FINE-TUNING CHANGES THE EMBEDDING DRAMATICALLY

Fig. 5 demonstrates this effect. The logits for query samples are far from those of their respective support samples and metric-based loss functions, e.g., those for prototypical networks (Snell et al., 2017) would have a high loss on this episode; indeed the accuracy after the support-based initialization is 64%. Logits for the query samples change dramatically during transductive fine-tuning and majority of the query samples cluster around their respective supports. The post transductive fine-tuning accuracy of this episode is 73.3%. This suggests that modifying the embedding using the query samples is crucial to obtaining good performance on new classes. This example also demonstrates that the support-based initialization is efficient, logits of the support samples are relatively unchanged during the transductive fine-tuning phase.

## C.2 Large vs. small backbones

The expressive power of the backbone plays an important role in the efficacy of fine-tuning. We observed that a WRN-16-4 architecture (2.7M parameters) performs worse than WRN-28-10 (36M parameters). The former obtains $63.28 \pm 0.68\%$ and $77.39 \pm 0.5\%$ accuracy on Mini-ImageNet and $69.04 \pm 0.69\%$ and $83.55 \pm 0.51\%$ accuracy on Tiered-ImageNet on 1-shot 5-way and 5-shot 5-way protocols respectively. While these numbers are comparable to those of state-of-the-art algorithms, they are lower than their counterparts for WRN-28-10 in Table 1. This suggests that a larger network is effective in learning richer features from the meta-training classes, and fine-tuning is effective in taking advantage of this to further improve performance on samples belonging to few-shot classes.

## C.3 Latency with a smaller backbones

The WRN-16-4 architecture (2.7M parameters) is much smaller than WRN-28-10 (36M parameters) and transductive fine-tuning on the former is much faster. As compared to our implementation of Snell et al. (2017) with the same backbone, WRN-16-4 is 20-70× slower (0.87 vs. 0.04 seconds for a query shot of 1, and 2.85 vs. 0.04 seconds for a query shot of 15) for the 1-shot 5-way scenario. Compare this to the computational complexity experiment in Section 4.3.

As discussed in Appendix C.2, the accuracy of WRN-16-4 is $63.28 \pm 0.68\%$ and $77.39 \pm 0.5\%$ for 1-shot 5-way and 5-shot 5-way on Mini-ImageNet respectively. As compared to this, our implementation of (Snell et al., 2017) using a WRN-16-4 backbone obtains $57.29 \pm 0.40\%$ and $75.34 \pm 0.32\%$ accuracies for the same settings respectively; the former number in particular is significantly worse than its transductive fine-tuning counterpart.

## C.4 Comparisons against backbones in the current literature

We include experiments using conv $(64)_{\times 4}$ and ResNet-12 in Table 3, in addition to WRN-28-10 in Section 4, in order to facilitate comparisons of the proposed baseline for different backbone architectures. Our results are comparable or better than existing results for a given backbone architecture, except for those in Liu et al. (2018b) who use a graph-based transduction algorithm, for conv $(64)_{\times 4}$ on Mini-ImageNet. In line with our goal for simplicity, we kept the hyper-parameters for pre-training and fine-tuning the same as the ones used for WRN-28-10 (cf. Sections 3 and 4). These results suggest that transductive fine-tuning is a sound baseline for a variety of backbone architectures.

## C.5 Using more meta-training classes

In Section 4.1 we observed that having more pre-training classes improves few-shot performance. But since we append a classifier on top of a pre-trained backbone and use the logits of the backbone as inputs to the classifier, a backbone pre-trained on more classes would also have more parameters as compared to one pre-trained on fewer classes. However, this difference is not large: WRN-28-10 for Mini-ImageNet has 0.03% more parameters for (train + val) as compared to (train). However, in order to facilitate a fair comparison, we ran an experiment where we use the features of the backbone, instead of the logits, as inputs to the classifier. By doing so, the number of parameters in the pre-trained backbone that are used for few-shot classification remain the same for both the (train) and (train + val) settings. For Mini-ImageNet, (train + val) obtains $64.20 \pm 0.65\%$ and $81.26 \pm 0.45\%$, and (train) obtains $62.55 \pm 0.65\%$ and $78.89 \pm 0.46\%$, for 1-shot 5-way and 5-shot 5-way respectively. These results corroborate the original statement that more pre-training classes improves few-shot performance.

## C.6 Using features of the backbone as input to the classifier

Instead of re-initializing the final fully-connected layer of the backbone to classify new classes, we simply append the classifier on top of it. We implemented the former, more common, approach and found that it achieves an accuracy of $64.20 \pm 0.65\%$ and $81.26 \pm 0.45\%$ for 1-shot 5-way and 5-shot 5-way respectively on Mini-ImageNet, while the accuracy on Tiered-ImageNet is $67.14 \pm$

Table 3: **Few-shot accuracies on benchmark datasets for 5-way few-shot episodes.** The notation conv $(64^k)_{\times 4}$ denotes a CNN with 4 layers and $64^k$ channels in the $k^{\text{th}}$ layer. The rows are grouped by the backbone architectures. Best results in each column and for a given backbone architecture are shown in bold. Results where the support-based initialization is better than or comparable to existing algorithms are denoted by [†]. The notation (train + val) indicates that the backbone was pre-trained on both training and validation sets of the datasets; the backbone is trained only on the training set otherwise. (Lee et al., 2019) uses a $1.25\times$ wider ResNet-12 which we denote as ResNet-12 [*].

| Algorithm | Architecture | Mini-ImageNet | | Tiered-ImageNet | | CIFAR-FS | | FC-100 | |
|---|---|---|---|---|---|---|---|---|---|
| | | 1-shot (%) | 5-shot (%) | 1-shot (%) | 5-shot (%) | 1-shot (%) | 5-shot (%) | 1-shot (%) | 5-shot (%) |
| MAML (Finn et al., 2017) | conv $(32)_{\times 4}$ | $48.70 \pm 1.84$ | $63.11 \pm 0.92$ | | | | | | |
| Matching networks (Vinyals et al., 2016) | conv $(64)_{\times 4}$ | $46.6$ | $60$ | | | | | | |
| LSTM meta-learner (Ravi & Larochelle, 2016) | conv $(64)_{\times 4}$ | $43.44 \pm 0.77$ | $60.60 \pm 0.71$ | | | | | | |
| Prototypical Networks (Snell et al., 2017) | conv $(64)_{\times 4}$ | $49.42 \pm 0.78$ | $68.20 \pm 0.66$ | | | | | | |
| Transductive Propagation (Liu et al., 2018b) | conv $(64)_{\times 4}$ | $\mathbf{55.51 \pm 0.86}$ | $\mathbf{69.86 \pm 0.65}$ | $\mathbf{59.91 \pm 0.94}$ | $73.30 \pm 0.75$ | | | | |
| Support-based initialization (train) | conv $(64)_{\times 4}$ | $50.69 \pm 0.63$ | $66.07 \pm 0.53$ | $58.42 \pm 0.69$ | $73.98 \pm 0.58^†$ | $\mathbf{61.77 \pm 0.73}$ | $76.40 \pm 0.54$ | $\mathbf{36.07 \pm 0.54}$ | $48.72 \pm 0.57$ |
| Fine-tuning (train) | conv $(64)_{\times 4}$ | $49.43 \pm 0.62$ | $66.42 \pm 0.53$ | $57.45 \pm 0.68$ | $73.96 \pm 0.56$ | $59.74 \pm 0.72$ | $76.37 \pm 0.53$ | $35.46 \pm 0.53$ | $49.43 \pm 0.57$ |
| Transductive fine-tuning (train) | conv $(64)_{\times 4}$ | $50.46 \pm 0.62$ | $66.68 \pm 0.52$ | $58.05 \pm 0.68$ | $\mathbf{74.24 \pm 0.56}$ | $61.73 \pm 0.72$ | $\mathbf{76.92 \pm 0.52}$ | $\mathbf{36.62 \pm 0.55}$ | $\mathbf{50.24 \pm 0.58}$ |
| R2D2 (Bertinetto et al., 2018) | conv $(96^k)_{\times 4}$ | $51.8 \pm 0.2$ | $68.4 \pm 0.2$ | | | $65.4 \pm 0.2$ | $79.4 \pm 0.2$ | | |
| TADAM (Oreshkin et al., 2018) | ResNet-12 | $58.5 \pm 0.3$ | $\mathbf{76.7 \pm 0.3}$ | | | | | $40.1 \pm 0.4$ | $\mathbf{56.1 \pm 0.4}$ |
| Transductive Propagation (Liu et al., 2018b) | ResNet-12 | $59.46$ | $75.64$ | | | | | | |
| Support-based initialization (train) | ResNet-12 | $54.21 \pm 0.64$ | $70.58 \pm 0.54$ | $66.39 \pm 0.73$ | $81.93 \pm 0.54$ | $65.69 \pm 0.72$ | $79.95 \pm 0.51$ | $35.51 \pm 0.53$ | $48.26 \pm 0.54$ |
| Fine-tuning (train) | ResNet-12 | $56.67 \pm 0.62$ | $74.80 \pm 0.51$ | $64.45 \pm 0.70$ | $\mathbf{83.59 \pm 0.51}$ | $64.66 \pm 0.73$ | $\mathbf{82.13 \pm 0.50}$ | $37.52 \pm 0.53$ | $55.39 \pm 0.57$ |
| Transductive fine-tuning (train) | ResNet-12 | $\mathbf{62.35 \pm 0.66}$ | $74.53 \pm 0.54$ | $\mathbf{68.41 \pm 0.73}$ | $83.41 \pm 0.52$ | $\mathbf{70.76 \pm 0.74}$ | $81.56 \pm 0.53$ | $\mathbf{41.89 \pm 0.59}$ | $54.96 \pm 0.55$ |
| MetaOpt SVM (Lee et al., 2019) | ResNet-12 [*] | $62.64 \pm 0.61$ | $78.63 \pm 0.46$ | $65.99 \pm 0.72$ | $81.56 \pm 0.53$ | $72.0 \pm 0.7$ | $84.2 \pm 0.5$ | $41.1 \pm 0.6$ | $55.5 \pm 0.6$ |
| Support-based initialization (train) | WRN-28-10 | $56.17 \pm 0.64$ | $73.31 \pm 0.53$ | $67.45 \pm 0.70$ | $82.88 \pm 0.53$ | $70.26 \pm 0.70$ | $83.82 \pm 0.49$ | $36.82 \pm 0.51$ | $49.72 \pm 0.55$ |
| Fine-tuning (train) | WRN-28-10 | $57.73 \pm 0.62$ | $\mathbf{78.17 \pm 0.49}$ | $66.58 \pm 0.70$ | $\mathbf{85.55 \pm 0.48}$ | $68.72 \pm 0.67$ | $\mathbf{86.11 \pm 0.47}$ | $38.25 \pm 0.52$ | $\mathbf{57.19 \pm 0.57}$ |
| Transductive fine-tuning (train) | WRN-28-10 | $\mathbf{65.73 \pm 0.68}$ | $78.40 \pm 0.52$ | $\mathbf{73.34 \pm 0.71}$ | $85.50 \pm 0.50$ | $\mathbf{76.58 \pm 0.68}$ | $85.79 \pm 0.50$ | $\mathbf{43.16 \pm 0.59}$ | $57.57 \pm 0.55$ |
| Support-based initialization (train + val) | conv $(64)_{\times 4}$ | $\mathbf{52.77 \pm 0.64}$ | $68.29 \pm 0.54$ | $\mathbf{59.08 \pm 0.70}$ | $74.62 \pm 0.57$ | $\mathbf{64.01 \pm 0.71}$ | $78.46 \pm 0.53$ | $40.25 \pm 0.56$ | $54.53 \pm 0.57$ |
| Fine-tuning (train + val) | conv $(64)_{\times 4}$ | $51.40 \pm 0.61$ | $68.58 \pm 0.52$ | $58.04 \pm 0.68$ | $74.48 \pm 0.56$ | $62.12 \pm 0.71$ | $77.98 \pm 0.52$ | $39.09 \pm 0.55$ | $54.83 \pm 0.55$ |
| Transductive fine-tuning (train + val) | conv $(64)_{\times 4}$ | $52.30 \pm 0.61$ | $\mathbf{68.78 \pm 0.53}$ | $58.81 \pm 0.69$ | $\mathbf{74.71 \pm 0.56}$ | $63.89 \pm 0.71$ | $\mathbf{78.48 \pm 0.52}$ | $\mathbf{40.33 \pm 0.56}$ | $\mathbf{55.60 \pm 0.56}$ |
| Support-based initialization (train + val) | ResNet-12 | $56.79 \pm 0.65$ | $72.94 \pm 0.55$ | $67.60 \pm 0.71$ | $83.09 \pm 0.53$ | $69.39 \pm 0.71$ | $83.27 \pm 0.50$ | $43.11 \pm 0.58$ | $58.16 \pm 0.57$ |
| Fine-tuning (train + val) | ResNet-12 | $58.64 \pm 0.64$ | $\mathbf{76.83 \pm 0.50}$ | $65.55 \pm 0.70$ | $\mathbf{84.51 \pm 0.50}$ | $68.11 \pm 0.70$ | $\mathbf{85.19 \pm 0.48}$ | $42.84 \pm 0.57$ | $\mathbf{63.10 \pm 0.57}$ |
| Transductive fine-tuning (train + val) | ResNet-12 | $\mathbf{64.50 \pm 0.68}$ | $76.92 \pm 0.55$ | $\mathbf{69.48 \pm 0.73}$ | $84.37 \pm 0.51$ | $\mathbf{74.35 \pm 0.71}$ | $84.57 \pm 0.53$ | $\mathbf{48.29 \pm 0.63}$ | $63.38 \pm 0.58$ |
| MetaOpt SVM (Lee et al., 2019) (train + val) | ResNet-12 [*] | $64.09 \pm 0.62$ | $80.00 \pm 0.45$ | $65.81 \pm 0.74$ | $81.75 \pm 0.53$ | $72.8 \pm 0.7$ | $85.0 \pm 0.5$ | $47.2 \pm 0.6$ | $62.5 \pm 0.6$ |
| Activation to Parameter (Qiao et al., 2018) (train + val) | WRN-28-10 | $59.60 \pm 0.41$ | $73.74 \pm 0.19$ | | | | | | |
| LEO (Rusu et al., 2018) (train + val) | WRN-28-10 | $61.76 \pm 0.08$ | $77.59 \pm 0.12$ | $66.33 \pm 0.05$ | $81.44 \pm 0.09$ | | | | |
| Support-based initialization (train + val) | WRN-28-10 | $58.47 \pm 0.66$ | $75.56 \pm 0.52$ | $67.34 \pm 0.69^†$ | $83.32 \pm 0.51^†$ | $72.14 \pm 0.69$ | $85.21 \pm 0.49$ | $45.08 \pm 0.61$ | $60.05 \pm 0.60$ |
| Fine-tuning (train + val) | WRN-28-10 | $59.62 \pm 0.66$ | $\mathbf{79.93 \pm 0.47}$ | $66.23 \pm 0.68$ | $\mathbf{86.08 \pm 0.47}$ | $70.07 \pm 0.67$ | $\mathbf{87.26 \pm 0.45}$ | $43.80 \pm 0.58$ | $64.40 \pm 0.58$ |
| Transductive fine-tuning (train + val) | WRN-28-10 | $\mathbf{68.11 \pm 0.69}$ | $80.36 \pm 0.50$ | $\mathbf{72.87 \pm 0.71}$ | $86.15 \pm 0.50$ | $\mathbf{78.36 \pm 0.70}$ | $87.54 \pm 0.49$ | $\mathbf{50.44 \pm 0.68}$ | $\mathbf{65.74 \pm 0.60}$ |

$0.74\%$ and $86.67 \pm 0.46\%$ for 1-shot 5-way and 5-shot 5-way respectively. These numbers are significantly lower for the 1-shot 5-way protocol on both datasets compared to their counterparts in Table 1. However, the 5-shot 5-way accuracy is marginally higher in this experiment than that in Table 1. As noted in Remark 2, logits of the backbone are well-clustered and that is why they work better for few-shot scenarios.

## C.7 Freezing the backbone restricts performance

The previous observation suggests that the network changes a lot in the fine-tuning phase. Freezing the backbone severely restricts the changes in the network to only changes to the classifier. As a consequence, the accuracy of freezing the backbone is $58.38 \pm 0.66$ % and $75.46 \pm 0.52$% on Mini-ImageNet and $67.06 \pm 0.69$% and $83.20 \pm 0.51$% on Tiered-ImageNet for 1-shot 5-way and 5-shot 5-way respectively. While the 1-shot 5-way accuracies are much lower than their counterparts in Table 1, the gap in the 5-shot 5-way scenario is smaller.

## C.8 Using mixup during pre-training

Mixup improves the few-shot accuracy by about 1%; the accuracy for WRN-28-10 trained without mixup is $67.06 \pm 0.71$% and $79.29 \pm 0.51$% on Mini-ImageNet for 1-shot 5-way and 5-shot 5-way respectively.

## C.9 More few-shot episodes

Fig. 1 suggests that the standard deviation of the accuracies achieved by few-shot algorithms is high. Considering this randomness, evaluations were done over 10,000 few-shot episodes as well. The accuracies on Mini-ImageNet are $67.77 \pm 0.21$ % and $80.24 \pm 0.16$ % and on Tiered-ImageNet are $72.36 \pm 0.23$ % and $85.70 \pm 0.16$ % for 1-shot 5-way and 5-shot 5-way respectively. The numbers are consistent with the ones for 1,000 few-shot episodes in Table 1, though the confidence intervals decreased as the number of episodes sampled increased.

## C.10 Evaluation on Meta-Dataset

Table 4: **Few-shot accuracies on Meta-Dataset:** Best results in each row are shown in bold. 600 few-shot episodes were used to compare to the results reported in Triantafillou et al. (2019).

| Dataset | Best performance in Triantafillou et al. (2019) | Transductive Fine-tuning | Rank for Transductive Fine-tuning (based on Triantafillou et al. (2019)) |
|---|---|---|---|
| ImageNet-1k (ILSVRC) | $51.01 \pm 1.05$ | $\mathbf{55.57 \pm 1.02}$ | 1 |
| Omniglot | $63.00 \pm 1.35$ | $\mathbf{79.59 \pm 0.98}$ | 1 |
| Aircraft | $\mathbf{68.69 \pm 1.26}$ | $67.26 \pm 0.98$ | 1.5 |
| Birds | $68.79 \pm 1.01$ | $\mathbf{74.26 \pm 0.82}$ | 1 |
| Textures | $69.05 \pm 0.90$ | $\mathbf{77.35 \pm 0.74}$ | 1 |
| VGG Flowers | $\mathbf{86.86 \pm 0.75}$ | $\mathbf{88.14 \pm 0.63}$ | 1.5 |
| Traffic Signs | $\mathbf{66.79 \pm 1.31}$ | $55.98 \pm 1.32$ | 2 |
| MSCOCO | $\mathbf{43.41 \pm 1.06}$ | $40.62 \pm 0.98$ | 2.5 |
| **Average Rank** | | | **1.4375** |

We ran experiments on Meta-Dataset (Triantafillou et al., 2019), and compared the performance of transductive fine-tuning for meta-training done on ImageNet-1k (ILSVRC) in Table 4. Transductive fine-tuning is better, most times significantly, than state-of-the-art on 6 out of 8 tasks in Meta-Dataset; its average rank across all tasks is 1.4375 (calculated using the results reported in Triantafillou et al. (2019)). The Fungi and Quick Draw datasets were not included because of issues with getting the data; the link to access the dataset for the former does not seem to work and the latter requires certain legal conditions which we are working on obtaining.

The few-shot episode sampling was done the same way as described in Triantafillou et al. (2019); except for the few-shot class sampling for ImageNet-1k (ILSVRC) and Omniglot, which was done uniformly over all few-shot classes (Triantafillou et al. (2019) use a hierarchical sampling technique to sample classes that are far from each other in the hierarchy, and hence easier to distinguish between). The hyper-parameters used for meta-training and few-shot fine-tuning are kept the same as the ones in Section 4 and are not tuned for these experiments.

## D  FREQUENTLY ASKED QUESTIONS

1. **Why has it not been noticed yet that this simple approach works so well?**

   Non-transductive fine-tuning as a baseline has been considered before (Vinyals et al., 2016; Chen et al., 2018). The fact that this is comparable to state-of-the-art has probably gone unnoticed because of the following reasons:

   - Given that there are only a few labeled support samples provided in the few-shot setting, initializing the classifier becomes important. The support-based initialization (cf. Section 3.1) motivated from the deep metric learning literature (Hu et al., 2015; Movshovitz-Attias et al., 2017; Qi et al., 2018; Gidaris & Komodakis, 2018) classifies support samples correctly (for a support shot of 1, this may not be true for higher support shots). This initialization, as opposed to initializing the weights of the classifier randomly, was critical to performance in our experiments.
   - In our experience, existing meta-training methods, both gradient-based ones and metric-based ones, are difficult to tune for larger architectures. We speculate that this is the reason a large part of the existing literature focuses on smaller backbone architectures. The few-shot learning literature has only recently started to move towards bigger backbone architectures (Oreshkin et al., 2018; Rusu et al., 2018). From Table 3 we see that non-tranductive fine-tuning gets better with a deeper backbone architecture. A similar observation was made by (Chen et al., 2018). The observation that we can use "simple" well-understood training techniques from standard supervised learning that scale up to large backbone architectures for few-shot classification is a key contribution of our paper.

   Transductive methods have recently started to become popular in the few-shot learning literature (Nichol et al., 2018; Liu et al., 2018a). Because of the scarcity of labeled support samples, it is crucial to make use of the unlabeled query samples in the few-shot regime.

   Our advocated baseline makes use of both a good initialization and transduction, relatively new in the few-shot learning literature, which makes this simplistic approach go unrecognized till now.

2. **Transductive fine-tuning works better than existing algorithms because of a big backbone architecture. One should compare on the same backbone architectures as the existing algorithms for a fair comparison.**

   The current literature is in the spirit of increasingly sophisticated approaches for modest performance gains, often with different architectures (cf. Table 1). This is why we set out to establish a baseline. Our simple baseline is comparable or better than existing approaches. The backbone we have used is common in the recent few-shot learning literature (Rusu et al., 2018; Qiao et al., 2018) (cf. Table 1). Additionally, we have included results on smaller common backbone architectures, namely conv $(64)_{\times 4}$ and ResNet-12 in Appendix C.4, and some additional experiments in Appendix C.2. These experiments suggest that transductive fine-tuning is a sound baseline for a variety of different backbone architectures. This indicates that we should take results on existing benchmarks with a grain of salt. Also see the response to question 1 above.

3. **There are missing entries in Tables 1 and 3. Is it still a fair comparison?**

   Tables 1 and 3 show all relevant published results by the original authors. Re-implementing existing algorithms to fill missing entries without access to original code is impractical and often yields results inferior to those published, which may be judged as unfair. The purpose of a benchmark is to enable others to test their method easily. This does not exist today due to myriad performance-critical design choices often not detailed in the papers. In fact, missing entries in the table indicate the inadequate state of the current literature. Our work enables benchmarking relative to a simple, systematic baseline.

4. **Fine-tuning for few-shot learning is not novel.**

   We do not claim novelty in this paper. Transductive fine-tuning is our advocated baseline for few-shot classification. It is a combination of different techniques that are not novel. Yet, it performs better than existing algorithms on all few-shot protocols with fixed hyper-parameters. We emphasize that this indicates the need to re-interpret existing results on benchmarks and re-evaluate the status quo in the literature.

5. **Transductive fine-tuning has a very high latency at inference time, this is not practical.**

   Our goal is to establish a systematic baseline for accuracy, which might help judge the accuracy of few-shot learning algorithms in the future. The question of test-time latency is indeed important but we have not focused on it in this paper. Appendix C.3 provides results using a smaller backbone where we see that the WRN-16-4 network is about 20-70x slower than metric-based approaches employing the same backbone while having significantly better accuracy. The latencies with WRN-28-10 are larger (see the computational complexity section in Section 4.3) but with a bigger advantage in terms of accuracy.

   There are other transductive methods used for few-shot classification (Nichol et al., 2018; Liu et al., 2018a), that are expected to be slow as well.

6. **Transductive fine-tuning does not make sense in the online setting when query samples are shown in a sequence.**

   Transductive fine-tuning can be performed even with a single test datum. Indeed, the network can specialize itself completely to classify this one datum. We explore a similar scenario in Section 4.3 and Fig. 2a, which discuss the performance of transductive fine-tuning with a query shot of 1 (this means 5 query samples one from each class for 5-way evaluation). Note that the loss function in (8) leverages multiple query samples when available. It does not require that the query samples be balanced in terms of their ground-truth classes. In particular, the loss function in (8) is well-defined even for a single test datum. For concerns about latency, see the question 5 above.

7. **Having transductive approaches will incentivize hacking the query set.**

   There are already published methods that use transductive methods (Nichol et al., 2018; Liu et al., 2018a), and it is a fundamental property of the transductive paradigm to be dependent on the query set, in addition to the support set. In order to prevent query set hacking, we will make the test episodes public which will enable consistent benchmarking, even for transductive methods.

8. **Why is having the same hyper-parameters for different few-shot protocols so important?**

   A practical few-shot learning algorithm should be able to handle any few-shot protocol. Having one model for each different scenario is unreasonable in the real-world, as the number of different scenarios is, in principle, infinite. Current algorithms do not handle this well. A single model which can handle any few-shot scenario is thus desirable.

9. **Is this over-fitting to the test datum?**

   No, label of the test datum is not used in the loss function.

10. **Can you give some intuition about the hardness metric? How did you come up with the formula?**

    The hardness metric is the clustering loss where the labeled support samples form the centers of the class-specific clusters. The special form, namely, $E_{(x,y)\in\mathcal{D}_q} \log \frac{1-p(y|x)}{p(y|x)}$ (cf. (9)) allows an interpretation of log-odds. We used this form because it is sensitive to the number of few-shot classes (cf. Fig. 3). Similar metrics, e.g., $E_{(x,y)\in\mathcal{D}_q} [-\log p(y|x)]$ can also be used but they come with a few caveats. Note that it is easier for $p(y|x)$ to be large for small way because the normalization constant in softmax has fewer terms. For large way, $p(y|x)$ could be smaller. This effect is better captured by our metric.

11. **How does Fig. 3 look for algorithm X, Y, Z?**

    We compared two algorithms in Fig. 3, namely transductive fine-tuning and support-based initialization. Section 4.4 and the caption of Fig. 3 explains how the former algorithm is better. We will consider adding comparisons to other algorithms to this plot in the future.

