# OpenReview forum: "A Baseline for Few-Shot Image Classification"
_ICLR.cc/2020/Conference — Accept (Poster)_

### Official Review · AnonReviewer2 · 2019-10-23
**Official Blind Review #2**

**Rating:** 6

**Review:**

This paper provided a baseline method for few-shot learning. It utilizes a simple but effective approach via a transductive fine-tuning. The experimental results on several benchmarks show the improvements over state-of-the-art approaches.

It is a comprehensive study of the methods and datasets in this domain. The motivation, experimental details and result analysis are clear to me. Overall, the paper is well written and the author is very transparent to show what they have.

The only drawback of this paper is it does not provide insight/explanation. How can the simple baseline work sowell? Is this because of some bias from the datasets? I also suggest that the author can try their method on some new dataset, like Meta-Dataset (Triantafillou et al. 2019).

I agreed with the author that the paper is not novel. However, I think the acceptance of the paper could benefit the community and I encourage the author can try this on some new benchmark. Therefore, I made my recommendation.

**Experience Assessment:**

I have published one or two papers in this area.

**Review Assessment: Checking Correctness Of Derivations And Theory:**

N/A

**Review Assessment: Checking Correctness Of Experiments:**

I carefully checked the experiments.

**Review Assessment: Thoroughness In Paper Reading:**

I read the paper at least twice and used my best judgement in assessing the paper.

---

> ### Author Response · Authors · 2019-11-15
> **Response to Review #2**
>
> We thank the reviewer for their feedback. Please also see our response to all the reviewers in the comment above.
>
> >>> How can the simple baseline work so well?
>
> We have included an answer to this question in Appendix D, question 1. The main reasons for the strong performance are:
> 1. It is critical to have efficient algorithms for adaptation in the case with few-labeled data. The metric-learning based initialization is important.
> 2. Not all existing algorithms use state-of-the-art backbone architectures, e.g,. the (conv-64)_{x4} network that is popular has only about 225,000 parameters (for comparison, LeNet for MNIST has about 130,000).
>
> >>> Is this because of some bias from the datasets?
>
> We do not believe our strong results are due to "bias" in the datasets. The hardness metric in Section 4.4 and Figure 3 shows that none of the datasets are unduly easy; their hardness is spread across the X-axis, only constrained by the size of the datasets themselves.
>
> >>> I also suggest that the author can try their method on some new dataset, like Meta-Dataset (Triantafillou et al. 2019).
>
> Thanks for this suggestion. We ran experiments on Meta-Dataset which we will add to the main paper.
>
> Transductive fine-tuning is better, most times significantly, than SoTA on 6/8 tasks in Meta-Dataset, the average rank across all tasks is 1.4375. We did not change hyper-parameters for transductive fine-tuning and kept them to the same values as our original submission. We could not find the link to the Fungi dataset, the original link does not seem to work anymore. Using the Quick Draw dataset requires us to accept certain legal conditions; we are working on getting the approval to use this dataset.
>
> +----------------------------+---------------------------+------------------------------------+-------------------------------------------------+
> |Dataset                       | Best performance  | Transductive Fine-tuning | Rank for Transductive Fine-Tuning |
> |                                     |  in Meta-Dataset    |                                               |           (based on Meta-Dataset)        |
> +----------------------------+---------------------------+------------------------------------+-------------------------------------------------+
> | ImageNet (ILSVRC)  |     51.01 +/- 1.05      |         55.57 +/- 1.02              |                                 1                            |
> | Omniglot                   |     63.00 +/- 1.35      |         79.59 +/- 0.98              |                                 1                            |
> | Aircraft                      |     68.69 +/- 1.26       |         67.26 +/- 0.98              |                               1.5                           |
> | Birds                          |     68.79 +/- 1.01       |         74.26 +/- 0.82              |                                 1                            |
> | Textures                    |     69.05 +/- 0.90       |         77.35 +/- 0.74              |                                 1                            |
> | VGG Flowers             |     86.86 +/- 0.75       |         88.14 +/- 0.63             |                               1.5                            |
> | Traffic Signs             |     66.79 +/- 1.31       |          55.98 +/- 1.32             |                                 2                             |
> | MSCOCO                   |     43.41 +/- 1.06       |          40.62 +/- 0.98             |                               2.5                           |
> +----------------------------+---------------------------+------------------------------------+-------------------------------------------------+
> | Average Rank                                                                                                 |                            1.4375                        |
> +----------------------------+---------------------------+------------------------------------+-------------------------------------------------+
>
> The original Meta-Dataset paper samples few-shot episodes for ImageNet and Omniglot by sampling classes that are far away from each other, this therefore creates easier episodes (easily distinguishable). We did not do this for our experiments and simply sampled the classes uniformly at random, which also creates harder episodes. An interesting thing to note above is that transductive fine-tuning has consistently lower standard error in the accuracy than the original results (for the same number of few-shot episodes).
>
> [Triantafillou et al.] Meta-Dataset: A Dataset of Datasets for Learning to Learn from Few Examples

---

### Official Review · AnonReviewer3 · 2019-10-23
**Official Blind Review #3**

**Rating:** 6

**Review:**

The authors propose a fine-tune-based few-shot classification baseline, which has been validated effectively on several datasets, including Mini-Imagenet, Tiered-Imagenet, CIFAR-FS, FC-100, and Imagenet-21k. In addition to the method, the authors also provide concrete experimental setting and new evaluation proposals.

1. The authors propose to use the logits instead of embedding as the main bridge between the pre-trained model and the meta-learning model. Does it mean we represent novel classes based on the properties of the meta-train classes? If so, does this method requires more meta-train classes to enrich the representation ability? How will the method perform when working on few-shot learning problems with a large distribution shift?

2. To make a fair comparison: instead of citing the values in the published papers directly and comparing different methods with different architectures, the authors should also apply the pre-trained model with the famous baselines, such as Matching Network, Prototypical Network, and MAML. Now there exists a very lap gap between the Matching Network values and the newly proposed one. For example, fine-tune the Matching Network on the pre-trained backbone in both train and train+val settings. Therefore, it is more clear to show the improvement of the proposed baseline models. Q/A 2-3 in appendix D do not fully solve this problem.

3. Considering the randomness of the sampled few-shot tasks, the authors can consider evaluating over more episodes (e.g., 10,000 trials) than 1000 in the paper.

4. It's better for the authors to emphasize and differentiate the transductive fine-tune and the inductive counterpart in the paper.

**Experience Assessment:**

I have published one or two papers in this area.

**Review Assessment: Checking Correctness Of Derivations And Theory:**

N/A

**Review Assessment: Checking Correctness Of Experiments:**

I carefully checked the experiments.

**Review Assessment: Thoroughness In Paper Reading:**

I read the paper at least twice and used my best judgement in assessing the paper.

---

> ### Author Response · Authors · 2019-11-15
> **Response to Review #3 (Part 1)**
>
> We thank the reviewer for their feedback. Please also see our response to all the reviewers in the comment above.
>
> >>> The authors propose to use the logits instead of embedding as the main bridge between the pre-trained model and the meta-learning model. Does it mean we represent novel classes based on the properties of the meta-train classes?
>
> We are not sure of the meaning of the first sentence: perhaps the reviewer means "bridge between the pre-trained model and the fine-tuned/adapted model". Yes, we use the logits instead of the features as inputs to few-shot classifier: Remark 2 and Appendix C.6 explain the rationale for doing so.
>
> >>> If so, does this method requires more meta-train classes to enrich the representation ability?
>
> No, this method does not require more meta-training classes. We use the same number of meta-training classes as all other methods for all benchmarks. Having more meta-training classes certainly helps; our accuracy for 5-way 5-shot testing on ImageNet 21K (7,491 meta-training classes) is as high as 95%. Appendix C.6 reports the performance when embeddings are used instead of the logits.
>
> >>>  How will the method perform when working on few-shot learning problems with a large distribution shift?
>
> Both logits and features are a property of the meta-training set, both may suffer from distribution shift. Fine-tuning adapts the network explicitly and safeguards against distribution shift, we therefore expect our method to retain its performance gains with large distribution shift. See also the results on Meta-Dataset (discussed in the comments for all reviewers and Appendix C.10)
>
> >>> To make a fair comparison: instead of citing the values in the published papers directly and comparing different methods with different architectures, the authors should also apply the pre-trained model with the famous baselines, such as Matching Network, Prototypical Network, and MAML. Now there exists a very lap gap between the Matching Network values and the newly proposed one. For example, fine-tune the Matching Network on the pre-trained backbone in both train and train+val settings. Therefore, it is more clear to show the improvement of the proposed baseline models. Q/A 2-3 in appendix D do not fully solve this problem.
>
> We are not sure we completely understand what the reviewer is saying here.
>
> If the reviewer means "run famous baselines with your backbone architecture".
> As the reviewer can appreciate, it is very difficult to reproduce these published results without access to original author’s source code, or run them for newer architectures. In particular, we have not been able to reproduce the results of MAML, or obtain good results with others, e.g., Prototypical Networks on new backbone architectures. Table 3 in Appendix C.6 includes results of transductive fine-tuning on the architectures of these above algorithms, where we have similar performance gains on these algorithms.
>
> If the reviewer instead means that we should pre-train our backbone with other algorithms.
> Pre-training the backbone using other meta-training approaches will defy our baseline effort: it will not only make the "baseline" as complicated as the existing algorithms but is also prohibitively difficult to do without access to the original author's source code. Our baseline is to do standard supervised learning (no episodic meta-training) and then fine-tune transductively. This method is very simple to implement and thus can be considered a "baseline".
>
> >>> Therefore, it is more clear to show the improvement of the proposed baseline models
>
> The main point of the paper is to devise a (simple) baseline, and show that a trivial form of meta-training surpasses state-of-the-art methods. This is a statement about the current methods and the evaluation benchmarks, rather than an attempt at creating a plausible state-of-the-art system.

---

> ### Author Response · Authors · 2019-11-15
> **Response to Review #3 (Part 2)**
>
> >>> Considering the randomness of the sampled few-shot tasks, the authors can consider evaluating over more episodes (e.g., 10,000 trials) than 1000 in the paper.
>
> We evaluated on 10,000 episodes for Mini-ImageNet and Tiered-ImageNet for 5-way, 1-shot and 5-shot test protocols. The numbers are consistent with our reported results. We will add this table to the main paper.
>
>
> +-------------------------+------------------------+----------------------+-----------------------+----------------------+
> |                                 |                    1-shot, 5-way                  |                   5-shot, 5-way                  |
> +                                 +------------------------+----------------------+-----------------------+----------------------+
> |                                 | 10,000 episodes | 1,000 episodes | 10,000 episodes | 1,000 episodes |
> +-------------------------+------------------------+----------------------+-----------------------+----------------------+
> | Mini-ImageNet    | 67.77 +/- 0.21       | 68.11 +/- 0.69    | 80.24 +/- 0.16     | 80.36 +/- 0.50    |
> | Tiered-ImageNet | 72.36 +/- 0.23       | 72.87 +/- 0.71   | 85.70 +/- 0.16     | 86.15 +/- 0.50     |
> +-------------------------+------------------------+----------------------+-----------------------+----------------------+
>
> >>> It's better for the authors to emphasize and differentiate the transductive fine-tune and the inductive counterpart in the paper.
> Thanks. We will expand upon Section 3.2 with an example on transductive learning, similar to Figure 2 in the paper "Transductive Inference for Text Classication using Support Vector Machines". We will also clarify the difference between transductive fine-tuning and fine-tuning (its inductive counterpart) in Section 4.1.
>
> [Thorsten Joachims, 99] Transductive Inference for Text Classication using Support Vector Machines

---

### Official Review · AnonReviewer1 · 2019-10-31
**Official Blind Review #1**

**Rating:** 6

**Review:**

This paper introduces a transductive learning baseline for few-shot image classification. The proposed approach includes a standard cross-entropy loss on the labeled support samples and a Shannon entropy loss on the unlabeled query samples. Despite its simplicity, the experimental results show that it can consistently outperform the state-of-the-art on four public few-shot datasets. In addition, they introduce a large-scale few-shot benchmark with 21K classes of ImageNet21K. Finally, they point out that accuracies from different episodes have high variance and develop another few-shot performance metric based on the hardness of each episode.

Positive comments:
1. The proposed transductive loss that minimizes entropy of query samples is novel in few-shot learning. Given limited labeled samples, finetuning with unlabeled query samples via proper loss is a good idea to tackle few-shot learning.
2. The evaluation is thorough. A significant number of few-shot methods are compared on 4 exisiting few-shot benchmarks. An additional large-scale benchmark is also introduced to facilitate  the few-shot learning research.
3. A novel evaluation metric is proposed to evaluate few-shot learning methods under different difficulties level. Although I am convinced by the importance of such metric, it is interesting to supplement the averaged accuracy because it tells how the methods work under easy and difficult classes.

Negative comments:
1. The folloing important reference of the Shannon entropy on unlabeled data is missing. In fact, I suggest the authors to extend Section 3.2 a bit more because this is the main technic contribution.
Semi-supervised Learningby Entropy Minimization.  Grandvalet et al. NIPS 2015
2. I am not convinced by the necessity of the proposed hardness metric. The main argument of this paper is that accuracies over episodes have high variance. But isn't it expected that different  episode can include samples with different difficulties, leading to high variance of accuracies? I do not think it is realistic to have one algorithm that achieves similar accuracies on both easy and difficult tasks. The authors also fail to evaluate different methods with the proposed metric and show if this metric makes the ranking of algorithms different. Moreover, I find Figure 3 hard to interpret because there are too much information in it, including different colors, a lot of markers and lines. Why is the range of hardness 1-3 for some datasets and 1-5 for other datasets? I believe writting of Section 4.4 could be further improved.

Overall, I think this paper has significant contributions of proposing a novel few-shot baseline that establishes a new state-of-the-art and would recommend weak accept.



**Experience Assessment:**

I have published one or two papers in this area.

**Review Assessment: Checking Correctness Of Derivations And Theory:**

I carefully checked the derivations and theory.

**Review Assessment: Checking Correctness Of Experiments:**

I carefully checked the experiments.

**Review Assessment: Thoroughness In Paper Reading:**

I read the paper thoroughly.

---

> ### Author Response · Authors · 2019-11-15
> **Response to Review #1**
>
> We thank the reviewer for their feedback. Please also see our response to all the reviewers in the comment above.
>
> >>> Add reference to "Semi-supervised Learningby Entropy Minimization.  Grandvalet et al. NIPS 2015"
>
> Thanks. We already have this reference in our draft (Section 2.1).
>
> >>> In fact, I suggest the authors to extend Section 3.2 a bit more because this is the main technical contribution.
>
> We agree. We will expand upon Section 3.2 with an example on transductive learning, similar to Figure 2 in the paper "Transductive Inference for Text Classication using Support Vector Machines".
>
> >>> The main argument of this paper is that accuracies over episodes have high variance. But isn't it expected that different episode can include samples with different difficulties, leading to high variance of accuracies? I do not think it is realistic to have one algorithm that achieves similar accuracies on both easy and difficult tasks.
>
> The fact that episodes have high variance is _not_ our main argument. Figure 1 simply seeks to demonstrate that the way we measure the performance in few-shot learning may be fallible because of high variance across episodes. Identifying this is one of our results although not the main result of the paper. This is important because the standard deviation being so high has never been reported in the literature before. We agree with the reviewer completely on their second point: it is unlikely that one single algorithm will have similar accuracies on both easy and difficult tasks.
>
> >>> I am not convinced by the necessity of the proposed hardness metric
>
> As your previous question (and our response) says, intuitively few-shot tasks can be of diverse difficulty. The hardness metric is our attempt at answering the question: "how does one characterize the difficulty of a few-shot task"? This contribution is important, and we believe necessary, for two reasons.
>
> 1. The current accepted procedure of reporting mean and standard error does not capture the diversity of few-shot tasks. The proposed metric allows sampling tasks of a specific hardness and measuring their accuracy. One may thereby report a histogram of the accuracies, even for single few-shot protocol, like we have done in Figure 3.
> 2. Current algorithms train different models with different hyper-parameters for different few-shot protocols. Doing so is detrimental to ascertaining real-world few-shot performance where we do not control the way and shot. The metric provides a way to measure the performance of an algorithm across multiple protocols. This is similar to using an ROC curve for ascertaining both Type I and Type II errors in standard supervised learning.
>
> Please also see our response to the next two, related, comments.
>
> >>> The authors also fail to evaluate different methods with the proposed metric and show if this metric makes the ranking of algorithms different.
>
> As the reviewer can appreciate, it is difficult to evaluate the previous algorithms on all these ways and shot without access to the original published models and source code of the authors. We have not been able to reproduce numbers of famous algorithms, e.g., MAML, or obtain good results with others, e.g., Prototypical Networks on new backbone architectures.
>
> We have compared two algorithms discussed in our paper, namely support-based initialization and transductive fine-tuning using this metric. The former is better across all test protocols except for ImageNet-21K, where both are comparable. This is a sanity check for the hardness metric.
>
> We are proposing that this is one way to measure hardness and report results systematically. It is not the only way: ascertaining the efficacy of this metric and comparisons to other metrics will be part of future work.
>
> >>> I find Figure 3 hard to interpret because there are too much information in it, including different colors, a lot of markers and lines…. I believe writing of Section 4.4 could be further improved.
>
> We will improve the clarity of Section 4.4.
>
> We agree that there is a lot of information in Figure 3. The caption however explains every part of the figure, colors, markers and the lines. The purpose of plotting the data in one figure is to demonstrate that the hardness is a valid metric for all the 5 datasets, all the different shots and ways, and two different algorithms. This is an ambitious goal but proposing a new evaluation metric demands being thorough.
>
> >>> Why is the range of hardness 1-3 for some datasets and 1-5 for other datasets?
>
> The fact that Mini-ImageNet, CIFAR-FS and FC-100 have hardness 1-3 indicates that they are easy. This is primarily due to their evaluation datasets having fewer classes. For bigger datasets we can go to higher ways (we tested up to 160-way for Tiered-ImageNet and ImageNet-21K) and the maximum hardness is almost 5.
>
> [Thorsten Joachims, 99] Transductive Inference for Text Classication using Support Vector Machines

---

### Author Response · Authors · 2019-11-15
**Response to all reviewers**

We thank the reviewers for their feedback. We first summarize our response and the results of additional suggested experiments here. We have responded to the concerns of the reviewers as individual comments below.

All the reviewers were in agreement that our method, transductive fine-tuning, is sound and effective. They also agree that the results of the paper have been validated effectively and thoroughly on several datasets along with a large-scale experiment on ImageNet-21K.

As suggested by Reviewer 2 we have added an additional experiment on Meta-Dataset, a summary of the results (full results in Appendix C.10 and individual comment) is:

Transductive fine-tuning is better, most times significantly, than SoTA on 6/8 tasks in Meta-Dataset, the average rank across all tasks is 1.4375. We did not change hyper-parameters for transductive fine-tuning and kept them to the same values as our original submission. We could not find the link to the Fungi dataset, the original link does not seem to work anymore. Using the Quick Draw dataset requires us to accept certain legal conditions; we are working on getting the approval to use this dataset.

The main concern of Reviewer 3 is:
>>> To make a fair comparison, the authors should apply pre-trained model with the famous baselines, such as Matching Network, Prototypical Network, and MAML

We are not sure we completely understand what the reviewer is saying here.

If the reviewer means "run famous baselines with your backbone architecture".
As the reviewer can appreciate, it is very difficult to reproduce these published results without access to original author’s source code, or run them for newer architectures. In particular, we have not been able to reproduce the results of MAML, or obtain good results with others, e.g., Prototypical Networks on new backbone architectures. Table 3 in Appendix C.6 includes results of transductive fine-tuning on the architectures of these above algorithms, where we have similar performance gains on these algorithms.

If the reviewer instead means that we should pre-train our backbone with other algorithms.
Pre-training the backbone using other meta-training approaches will defy our baseline effort: it will not only make the "baseline" as complicated as the existing algorithms but is also prohibitively difficult to do without access to the original author's source code. Our baseline is to do standard supervised learning (no episodic meta-training) and then fine-tune transductively.

In a field that may be crucial for ushering in personalized machine learning, we believe our work is essential to ascertain the empirical performance of current algorithms and valuable to the community.

---

### Public Comment · ~Jackie_Cheung1 · 2020-05-01
**is it fair to compare Transductive setting vs inductive setting?**

hi. thanks for the nice baseline provided in this paper.
I have a few questions regarding your reported performance.  I have gone through your paper and think that your proposed Transductive fine-tuning is based on the transductive setting/ semi-supervised setting.  In this setting, the prediction of a query sample is based not only on the support images (training images) but also on many other unlabeled query images.
On the other hand,  many compared methods in Table1 is based on the inductive setting, where the prediction of individual query images is solely based on the support images without any other unlabeled data.  As far as I know, the performance gap between the benchmarks in these two settings is not small.  For example, in this work
https://arxiv.org/pdf/1911.06045.pdf,
the performance reaches 78% for 1-shot 5-way mini magnet.
Would it be better to indicate this more clearly or to make comparisons in different tables?

---

> ### Author Response · Authors · 2020-10-18
> **response**
>
> Thanks for your comment. Yes, the transductive setting and the inductive setting are different; accuracy of these methods should not be compared directly. We also show very strong results with non-transductive fine-tuning in Table 1 and clearly indicate in the narrative the difference between the two.
>
> Let us note that semi-supervised setting is a different than (it is a subset of) transductive learning. Transductive learning is powerful because while inductive learning seeks to achieve accurate predictions over the entire distribution of test data, transductive is about achieving accurate predictions only on a few particular samples of test data; in the few-shot learning problem this is the query shot. Transduction is particularly suited to problem settings when one is interested in getting accurate predictions only on the query samples of a particular episode. Semi-supervised learning is therefore a particular technique for implementing transduction.

---

### Public Comment · ~Mayank_Lunayach1 · 2020-10-18
**Source code**

Congrats on a fantastic work! When would the source code for the paper be released?

---

> ### Author Response · Authors · 2020-10-18
> **please write to us and we will be happy to help**
>
> Thank you for your comment. The code is going through internal reviews before it can be released. If you write to all the authors, we will be happy to guide you on your implementation over email.

---

### Decision · Program_Chairs · 2019-12-19

**Decision:**

Accept (Poster)

**Comment:**

This paper introduces a simple baseline for few-shot image classification in the transductive setting, which includes a standard cross-entropy loss on the labeled support samples and a conditional entropy loss on the unlabeled query samples.

Both losses are known in the literature (the seminal work of entropy minimization by Bengio should be cited properly). However, reviewers are positive about this paper, acknowledging the significant contributions of a novel few-shot baseline that establishes a new state-of-the-art on well-known public few-shot datasets as well as on the introduced large-scale benchmark ImageNet21K. The comprehensive study of the methods and datasets in this domain will benefit the research practices in this area.

Therefore, I make an acceptance recommendation.